# Certifiably Adversarially Robust Detection of Out-of-Distribution Data

**Julian Bitterwolf**
University of Tübingen

**Alexander Meinke**
University of Tübingen

**Matthias Hein**
University of Tübingen

## Abstract

Deep neural networks are known to be overconfident when applied to out-of-distribution (OOD) inputs which clearly do not belong to any class. This is a problem in safety-critical applications since a reliable assessment of the uncertainty of a classifier is a key property, allowing the system to trigger human intervention or to transfer into a safe state. In this paper, we aim for certifiable worst case guarantees for OOD detection by enforcing not only low confidence at the OOD point but also in an $l_\infty$-ball around it. For this purpose, we use interval bound propagation (IBP) to upper bound the maximal confidence in the $l_\infty$-ball and minimize this upper bound during training time. We show that non-trivial bounds on the confidence for OOD data generalizing beyond the OOD dataset seen at training time are possible. Moreover, in contrast to certified adversarial robustness which typically comes with significant loss in prediction performance, certified guarantees for worst case OOD detection are possible without much loss in accuracy.

## 1 Introduction

Deep neural networks are the state-of-the-art in many application areas. Nevertheless it is still a major concern to use deep learning in safety-critical systems, e.g. medical diagnosis or self-driving cars, since it has been shown that deep learning classifiers suffer from a number of unexpected failure modes, such as low robustness to natural perturbations [12, 17], overconfident predictions [31, 14, 18, 16] as well as adversarial vulnerabilities [36]. For safety critical applications, empirical checks are not sufficient in order to trust a deep learning system in a high-stakes decision. Thus provable guarantees on the behavior of a deep learning system are needed.

One property that one expects from a robust classifier is that it should *not* make highly confident predictions on data that is very different from the training data. However, ReLU networks have been shown to be provably overconfident far away from the training data [16]. This is a big problem as (guaranteed) low confidence of a classifier when it operates out of its training domain can be used to trigger human intervention or to let the system try to achieve a safe state when it "detects" that it is applied outside of its specification. Several approaches to the out-of-distribution (OOD) detection task have been studied [18, 25, 23, 24, 16]. The current state-of-the-art performance of OOD detection in image classification is achieved by enforcing low confidence on a large training set of natural images that is considered as out-distribution [19, 28].

Deep neural networks are also notoriously susceptible to small adversarial perturbations in the input [36, 4] which change the decision of a classifier. Research so far has concentrated on adversarial robustness around the in-distribution. Several empirical defenses have been proposed but many could be broken again [8, 3, 1]. Adversarial training and variations [27, 42] perform well empirically, but typically no robustness guarantees can be given. Certified adversarial robustness has been achieved by explicit computation of robustness certificates [15, 38, 33, 29, 13] and randomized smoothing [6].

Adversarial changes to generate high confidence predictions on the out-distribution have received much less attention although it has been shown early on that they can be used to fool a classifier

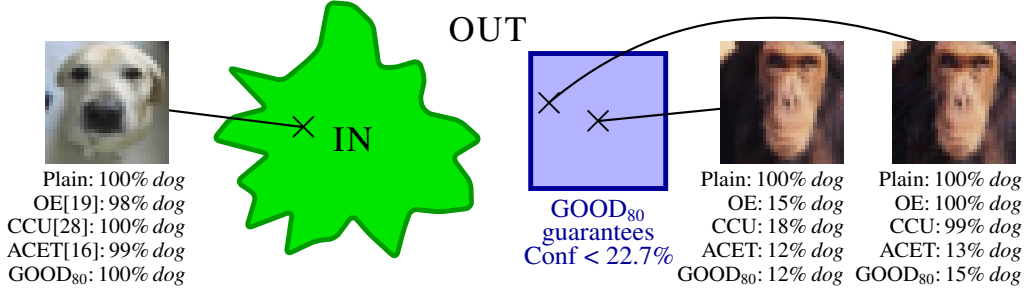

Figure 1: **Overconfident predictions on out-distribution inputs. Left:** On the in-distribution CIFAR-10 all methods have similar high confidence on the image of a *dog*. **Middle:** For the out-distribution image of a *chimpanzee* from CIFAR-100 the plain model is overconfident while an out-distribution aware method like Outlier Exposure (OE) [19] produces low confidence. **Right:** When maximizing the confidence inside the $l_\infty$-ball of radius $0.01$ around the *chimpanzee* image (for the OE model), OE as well as CCU become overconfident (right image). ACET and our $GOOD_{80}$ perform well in having empirical low confidence, but only $GOOD_{80}$ guarantees that the confidence in the $l_\infty$-ball of radius $0.01$ around the *chimpanzee* image (middle image) is less than 22.7% for any class (note that $10\%$ corresponds to maximal uncertainty as CIFAR-10 has 10 classes).

[31, 34, 35]. Thus, even if a classifier consistently manages to identify samples as not belonging to the in-distribution, it might still assign very high confidence to only marginally perturbed samples from the out-distribution, see Figure 1. A first empirical defense using a type of adversarial training for OOD detection has been proposed in [16]. However, up to our knowledge in the area of certified out-of-distribution detection the only robustness guarantees for OOD were given in [28], where a density estimator for in- and out-distribution is integrated into the predictive uncertainty of the neural network, which allows them to guarantee that far away from the training data the confidence of the neural network becomes uniform over the classes. Moreover, they provide guarantees on the maximal confidence attained on $l_2$-type balls around uniform noise. However, this technique is not able to provide meaningful guarantees around points which are similar or even close to the in-distribution data and, as we will show, provide only weak guarantees against $l_\infty$-adversaries.

In this work we aim to provide worst-case OOD guarantees not only for noise but also for images from related but different image classification tasks. For this purpose we use the techniques from interval bound propagation (IBP) [13] to derive a provable upper bound on the maximal confidence of the classifier in an $l_\infty$-ball of radius $\epsilon$ around a given point. By minimizing this bound on the out-distribution using our training scheme GOOD (Guaranteed Out-Of-distribution Detection) we arrive at the first models which have guaranteed low confidence even on image classification tasks related to the original one; e.g., we get state-of-the-art results on separating letters from EMNIST from digits in MNIST even though the digit classifier has never seen any images of letters at training time. In particular, the guarantees for the training out-distribution generalize to other out-distribution datasets. In contrast to classifiers which have certified adversarial robustness on the in-distribution, GOOD has the desirable property to achieve provable guarantees for OOD detection with almost no loss in accuracy on the in-distribution task even on datasets like CIFAR-10.

## 2 Out-of-distribution detection: setup and baselines

Let $f : \mathbb{R}^d \to \mathbb{R}^K$ be a feedforward neural network (DNN) with a last linear layer where $d$ is the input dimension and $K$ the number of classes. In all experiments below we use the ReLU activation function. The logits of $f(x)$ for $x \in \mathbb{R}^d$ are transformed via the softmax function into a probability distribution $p(x)$ over the classes with:

$$p_k(x) := \frac{e^{f_k(x)}}{\sum_{l=1}^{K} e^{f_l(x)}} \quad \text{for } k = 1, \ldots, K. \tag{1}$$

By $\mathrm{Conf}_f(x) = \max_{k=1,\ldots,K} p_k(x)$ we define the confidence of the classifier $f$ in the prediction $\mathrm{argmax}_{k=1,\ldots,K} p_k(x)$ at $x$.

The general goal of OOD detection is to construct a feature that can reliably separate the in-distribution from all inputs which clearly do not belong to the in-distribution task, especially inputs from regions which have zero probability under the in-distribution. One typical criterion to measure OOD detection performance is to use $\mathrm{Conf}_f(x)$ as a feature and compute the AUC of in- versus out-distribution (how well are confidences of in- and out-distribution separated). We discuss a proper conservative measurement of the AUC in case of indistinguishable confidence values, e.g. due to numerical precision, in Appendix C.

As baselines and motivation for our provable approach we use the OOD detection methods Outlier Exposure (OE) [19] and Confidence Enhancing Data Augmentation (CEDA) [16], which use as objective for training

$$\frac{1}{N}\sum_{i=1}^{N}\mathcal{L}_{\mathrm{CE}}(x_i^{\mathrm{IN}}, y_i^{\mathrm{IN}}) + \frac{\kappa}{M}\sum_{j=1}^{M}\mathcal{L}_{\mathrm{OUT}}(x_j^{\mathrm{OUT}})\,, \tag{2}$$

where $\left\{(x_i^{\mathrm{IN}}, y_i^{\mathrm{IN}}) \mid 1 \leq i \leq N\right\}$ is the in-distribution training set, $\left\{x_j^{\mathrm{OUT}} \mid 1 \leq j \leq M\right\}$ the out-distribution training set, and $\mathcal{L}_{\mathrm{CE}}$ the cross-entropy loss. The hyper-parameter $\kappa$ determines the relative magnitude of the two loss terms and is usually chosen to be one [19, 28, 16]. OE and CEDA differ in the choice of the loss $\mathcal{L}_{\mathrm{OUT}}$ for the out-distribution where OE uses the cross-entropy loss between $p(x_j^{\mathrm{OUT}})$ and the uniform distribution and CEDA uses $\log \mathrm{Conf}_f(x_j^{\mathrm{OUT}})$. Note that both the CEDA and OE loss attain their global minimum when $p(x)$ is the uniform distribution. Their difference is typically minor in practice. An important question is the choice of the out-distribution. For general image classification, it makes sense to use an out-distribution which encompasses basically any possible image one could ever see at test time and thus the set of all natural images is a good out-distribution; following [19] we use the 80 Million Tiny Images dataset [37] as a proxy for that.

While OE and CEDA yield state-of-the-art OOD detection performance for image classification tasks when used together with the 80M Tiny Images dataset as out-distribution, they are, similarly to normal classifiers, vulnerable to adversarial manipulation of the out-distribution images where the attack is trying to maximize the confidence in this scenario [28]. Thus [16] proposed Adversarial Confidence Enhanced Training (ACET) which replaces the CEDA loss with $\max_{\|\hat{x}-x_j^{\mathrm{OUT}}\|_\infty \leq \epsilon} \log \mathrm{Conf}_f(\hat{x})$ and can be seen as adversarial training on the out-distribution for an $l_\infty$-threat model. However, as for to adversarial training on the in-distribution [27] this does not yield any guarantees for out-distribution detection. In the next section we discuss how to use interval-bound-propagation (IBP) to get guaranteed OOD detection performance in an $l_\infty$-neighborhood of every out-distribution input.

## 3 Provable guarantees for out-of-distribution detection

Our goal is to minimize the confidence of the classifier not only on the out-distribution images themselves but in a whole neighborhood around them. For this purpose, we first derive bounds on the maximal confidence on some $l_\infty$-ball around a given point. In certified adversarial robustness, IBP [13] currently leads to the best guarantees for deterministic classifiers under the $l_\infty$-threat model. While other methods for deriving guarantees yield tighter bounds [38, 29], they are not easily scalable and, when optimized, the bounds given by IBP have been shown to be very tight [13].

**IBP.** Interval bound propagation [13] provides entrywise lower and upper bounds $\underline{z_k}^\epsilon$ resp. $\overline{z_k}^\epsilon$ for the output $z_k$ of the $k$-th layer of a neural network given that the input $x$ is varied in the $l_\infty$-ball of radius $\epsilon$. Let $\sigma : \mathbb{R} \to \mathbb{R}$ be a monotonically increasing activation function e.g. we use the ReLU function $\sigma(x) = \max\{0, x\}$ in the paper. We set $z_0 = x$ and $\underline{z_0}_\epsilon = x - \epsilon \cdot \mathbf{1}$ and $\overline{z_0}^\epsilon = x + \epsilon \cdot \mathbf{1}$ ($\mathbf{1}$ is the vector of all ones). If the $k$-th layer is linear (fully connected, convolutional, residual etc.) with weight matrix $W_k$, one gets upper and lower bounds of the next layer via forward propagation:

$$\overline{z_k}^\epsilon = \sigma\big(\max(W_k, 0) \cdot \overline{z_{k-1}}^\epsilon + \min(W_k, 0) \cdot \underline{z_{k-1}}_\epsilon + b_k\big)$$
$$\underline{z_k}_\epsilon = \sigma\big(\min(W_k, 0) \cdot \overline{z_{k-1}}^\epsilon + \max(W_k, 0) \cdot \underline{z_{k-1}}_\epsilon + b_k\big)\,, \tag{3}$$

where the $\min/\max$ expressions are taken componentwise and the activation function $\sigma$ is applied componentwise as well. Note that the derivation in [13] is slightly different, but the bounds are the same. The forward propagation of the bounds is of similar nature as a standard forward pass and back-propagation w.r.t. the weights is relatively straightforward.

**Upper bound on the confidence in terms of the logits.** The log confidence of the model at $x$ can be written as

$$\log \mathrm{Conf}(x) = \max_{k=1,\ldots,K} \log \frac{e^{f_k(x)}}{\sum_{l=1}^{K} e^{f_l(x)}} = \max_{k=1,\ldots,K} -\log \sum_{l=1}^{K} e^{f_l(x)-f_k(x)}. \tag{4}$$

We assume that the last layer is affine: $f(x) = W_L \cdot z_{L-1}(x) + b_L$, where $L$ is the number of layers of the network. We calculate the upper bounds of all $K^2$ logit differences as:

$$\begin{aligned}
\max_{\|\hat{x}-x\|_\infty \leq \epsilon} f_k(\hat{x}) - f_l(\hat{x}) &= \max_{\|\hat{x}-x\|_\infty \leq \epsilon} W_{L,k} \cdot z_{L-1}(\hat{x}) + b_{L,k} - W_{L,l} \cdot z_{L-1}(\hat{x}) - b_{L,l} \\
&= \max_{\|\hat{x}-x\|_\infty \leq \epsilon} (W_{L,k} - W_{L,l}) \cdot z_{L-1}(\hat{x}) + b_{L,k} - b_{L,l} \\
&\leq \max(W_{L,k} - W_{L,l}, 0) \cdot \overline{z_{L-1}(x)}^\epsilon \\
&\quad + \min(W_{L,k} - W_{L,l}, 0) \cdot \underline{z_{L-1}(x)}_\epsilon + b_{L,k} - b_{L,l} \\
&=: \overline{f_k(x) - f_l(x)}^\epsilon,
\end{aligned} \tag{5}$$

where $W_{L,k}$ denotes the $k$-th row of $W_L$ and $b_{L,k}$ is the $k$-th component of $b_L$. Note that this upper bound of the logit difference can be negative and is zero for $l = k$. Using this upper bound on the logit difference in Equation (4), we obtain an upper bound on the log confidence:

$$\max_{\|\hat{x}-x\|_\infty \leq \epsilon} \log \mathrm{Conf}(\hat{x}) \leq \max_{k=1,\ldots,K} -\log \sum_{l=1}^{K} e^{-(\overline{f_k(x)-f_l(x)}^\epsilon)} \tag{6}$$

We use the bound in (6) to evaluate the guarantees on the confidences for given out-distribution datasets. However, minimizing it directly during training leads to numerical problems, especially at the beginning of training, when the upper bounds $\overline{f_k(x) - f_l(x)}^\epsilon$ are very large for $l \neq k$ , which makes training numerically infeasible. Instead, we rather upper bound the log confidence again by bounding the sum inside the negative log from below with $K$ times its lowest term:

$$\begin{aligned}
\max_{k=1,\ldots,K} -\log \sum_{l=1}^{K} e^{-(\overline{f_k(x)-f_l(x)}^\epsilon)} &\leq \max_{k=1,\ldots,K} -\log \left( K \cdot \min_{l=1,\ldots,K} e^{-(\overline{f_k(x)-f_l(x)}^\epsilon)} \right) \\
&= \max_{k,l=1,\ldots,K} \overline{f_k(x) - f_l(x)}^\epsilon - \log K
\end{aligned} \tag{7}$$

While this bound can considerably differ from the potentially tighter bound of Equation (6), it is often quite close as one term in the sum dominates the others. Moreover, both bounds have the same global minimum when all logits are equal over the $l_\infty$-ball. We omit the constant $\log K$ in the following as it does not matter for training.

The direct minimization of the upper bound in (7) is still difficult, in particular for more challenging in-distribution datasets like SVHN and CIFAR-10, as the bound $\max_{k,l=1,\ldots,K} \overline{f_k(x) - f_l(x)}^\epsilon$ can be several orders of magnitude larger than the in-distribution loss. Therefore, we use the logarithm of this quantity. However, we also want to have a more fine-grained optimization when the upper bound becomes small in the later stage of the training. Thus we define the Confidence Upper Bound loss $\mathcal{L}_{\mathrm{CUB}}$ for an OOD input as

$$\mathcal{L}_{\mathrm{CUB}}(x; \epsilon) := \log \left( \frac{\left( \max_{k,l=1,\ldots,K} \overline{f_k(x) - f_l(x)}^\epsilon \right)^2}{2} + 1 \right). \tag{8}$$

Note that $\log(\frac{a^2}{2} + 1) \approx \frac{a^2}{2}$ for small $a$ and thus we achieve the more fine-grained optimization with an $l_2$-type of loss in the later stages of training which tries to get all upper bounds small. The overall objective of **fully applied Guaranteed OOD Detection training (GOOD$_{100}$)** is the minimization of

$$\frac{1}{N} \sum_{i=1}^{N} \mathcal{L}_{\mathrm{CE}}(x_i^{\mathrm{IN}}, y_i^{\mathrm{IN}}) + \frac{\kappa}{M} \sum_{j=1}^{M} \mathcal{L}_{\mathrm{CUB}}(x_j^{\mathrm{OUT}}; \epsilon) , \tag{9}$$

where $\left\{ (x_i^{\mathrm{IN}}, y_i^{\mathrm{IN}}) \mid 1 \leq i \leq N \right\}$ is the in-distribution training set and $\left\{ x_j^{\mathrm{OUT}} \mid 1 \leq j \leq M \right\}$ the out-distribution. The hyper-parameter $\kappa$ determines the relative magnitude of the two loss terms. During training we slowly increase this value and $\epsilon$ in order to further stabilize the training with GOOD.

**Quantile-GOOD: trade-off between clean and guaranteed AUC.** Training models by minimizing (9) means that the classifier gets severely punished if *any* training OOD input receives a high confidence upper bound. If OOD inputs exist to which the classifier already assigns high confidence without even considering the worst case, e.g. as these inputs share features with the in-distribution, it makes little sense to enforce low confidence guarantees. Later in the experiments we show that for difficult tasks like CIFAR-10 this can happen. In such cases the normal AUC for OOD detection gets worse as the high loss of the out-distribution part effectively leads to low confidence on a significant part of the in-distribution which is clearly undesirable.

Hence, for OOD inputs $x$ which are not clearly distinguishable from the in-distribution, it is preferable to just have the "normal" loss $\mathcal{L}_{\mathrm{CUB}}(x_j^{\mathrm{OUT}}; 0)$ without considering the worst case. We realize this by enforcing the loss with the guaranteed upper bounds on the confidence just on some quantile of the easier OOD inputs, namely the ones with the lowest guaranteed out-distribution loss $\mathcal{L}_{\mathrm{CUB}}(x; \epsilon)$. We first order the OOD training set by the potential loss $\mathcal{L}_{\mathrm{CUB}}(x; \epsilon)$ of each sample in ascending order $\pi$, that is $\mathcal{L}_{\mathrm{CUB}}(x_{\pi_1}^{\mathrm{OUT}}) \leq \mathcal{L}_{\mathrm{CUB}}(x_{\pi_2}^{\mathrm{OUT}}) \leq \ldots \leq \mathcal{L}_{\mathrm{CUB}}(x_{\pi_M}^{\mathrm{OUT}})$. We then apply the loss $\mathcal{L}_{\mathrm{CUB}}(x; \epsilon)$ to the lower quantile $q$ of the points (the ones with the smallest loss $\mathcal{L}_{\mathrm{CUB}}(x; \epsilon)$) and take $\mathcal{L}_{\mathrm{CUB}}(x; 0)$ for the remaining samples, which means no worst-case guarantees on the confidence are enforced:

$$\frac{1}{N}\sum_{i=1}^{N}\mathcal{L}_{\mathrm{CE}}(x_i^{\mathrm{IN}}, y_i^{\mathrm{IN}}) + \frac{\kappa}{M}\sum_{j=1}^{\lfloor q \cdot M \rfloor}\mathcal{L}_{\mathrm{CUB}}(x_{\pi_j}^{\mathrm{OUT}}; \epsilon) + \frac{\kappa}{M}\sum_{j=\lfloor q \cdot M \rfloor + 1}^{M}\mathcal{L}_{\mathrm{CUB}}(x_{\pi_j}^{\mathrm{OUT}}; 0) \,. \tag{10}$$

During training we do this ordering on the part of each batch consisting of out-distribution images. On CIFAR-10, where the out-distribution dataset 80M Tiny Images is closer to the in-distribution, the quantile GOOD-loss allows us to choose the trade-off between clean and guaranteed AUC for OOD detection, similar to the trade-off between clean and robust accuracy in adversarial robustness.

# 4 Experiments

We provide experimental results for image recognition tasks with MNIST [22], SVHN [30] and CIFAR-10 [21] as in-distribution datasets. We first discuss the training details, hyperparameters and evaluation before we present the results of GOOD and competing methods. Code is available under `https://gitlab.com/Bitterwolf/GOOD`.

## 4.1 Model architectures, training procedure and evaluation

**Model architectures and data augmentation.** For all experiments, we use deep convolutional neural networks consisting of convolutional, affine and ReLU layers. For MNIST, we use the large architecture from [13], and for SVHN and CIFAR-10 a similar but deeper and wider model. The layer structure is laid out in Table 2 in the appendix. Data augmentation is applied to both in- and out-distribution images during training. For MNIST we use random crops to size $28 \times 28$ with padding 4 and for SVHN and CIFAR-10 random crops with padding 4 as well as the quite aggressive augmentation AutoAugment [9]. Additionally, we apply random horizontal flips for CIFAR-10.

**GOOD training procedure.** As it is the case with IBP training [13] for certified adversarial robustness, we have observed that the inclusion of IBP bounds can make the training unstable or cause it to fail completely. This can happen for our GOOD training despite the logarithmic damping in the $\mathcal{L}_{\mathrm{CUB}}$ loss in (8). Thus, in order to further stabilize the training similar to [13], we use linear ramp up schedules for $\epsilon$ and $\kappa$, which are detailed in Appendix D. As radii for the $l_\infty$-perturbation model on the out-distribution we use $\epsilon = 0.3$ for MNIST, $\epsilon = 0.03$ for SVHN and $\epsilon = 0.01$ for CIFAR-10 (note that $0.01 > \frac{2}{255} \approx 0.0078$). The chosen $\epsilon = 0.01$ for CIFAR-10 is so small that the changes are hardly visible (see Figure 1). As parameter $\kappa$ for the trade-off between cross-entropy loss and the GOOD regularizer in (9) and (10), we set $\kappa = 0.3$ for MNIST and $\kappa = 1$ for SVHN and CIFAR-10.

In order to explore the potential trade-off between the separation of in- and out-distribution for clean and perturbed out-distribution inputs (clean AUCs vs guaranteed AUCs - see below), we train GOOD models for different quantiles $q \in [0, 1]$ in (10) which we denote as $\mathrm{GOOD}_Q$ in the following. Here, $Q = 100q$ is the percentage of out-distribution training samples for which we minimize the guaranteed upper bounds on the confidence of the neural network in the $l_\infty$-ball of radius $\epsilon$ around the out-distribution point during training. Note that $\mathrm{GOOD}_{100}$ corresponds to (9) where we minimize the guaranteed upper bound on the worst-case confidence for all out-distribution samples, whereas

GOOD$_0$ can be seen as a variant of OE or CEDA. A training batch consists of 128 in- and 128 out-distribution samples. Examples of OOD training batches with the employed augmentation and their quantile splits for a GOOD$_{60}$ model are shown in Table 3 in the appendix.

For the training out-distribution, we use 80 Million Tiny Images (80M) [37], which is a large collection of natural images associated to nouns in wordnet [11]. All methods get the same out-distribution for training and we are *neither* training *nor* adapting hyperparameters for each OOD dataset separately as in some previous work. Since CIFAR-10 and CIFAR-100 are subsets of 80M, we follow [19] and filter them out. As can be seen in the example batches in Table 3, even this reduced dataset still contains images from CIFAR-10 classes, which explains why our quantile-based loss is essential for good performance on CIFAR-10. We take a subset of 50 million images as OOD training set. Since the size of the training set of the in-distribution datasets (MNIST: 60,000; SVHN: 73,257; CIFAR-10: 50000) is small compared to 50 million, typically an OOD image appears only once during training.

**Evaluation.** For each method, we compute the test accuracy on the in-distribution task, and for various out-distribution datasets (not seen during training) we report the area under the receiver operating characteristic curve (AUC) as a measure for the separation of in- from out-distribution samples based on the predicted confidences on the test sets. As OOD evaluation sets we use FashionMNIST [39], the Letters of EMNIST [5], grayscale CIFAR-10, and Uniform Noise for MNIST, and CIFAR-100 [21], CIFAR-10/SVHN, LSUN Classroom [40], and Uniform Noise for SVHN/CIFAR-10. Further evaluation on other OOD datasets can be found in Appendix H.

We are particularly interested in the worst case OOD detection performance of all methods under the $l_\infty$-perturbation model for the out-distribution. For this purpose, we compute the **adversarial AUC (AAUC)** and the **guaranteed AUC (GAUC)**. These AUCs are based on the maximal confidence in the $l_\infty$-ball of radius $\epsilon$ around each out-distribution image. For the adversarial AUC, we compute a lower bound on the maximal confidence in the $l_\infty$-ball by using Auto-PGD [8] for maximizing the confidence of the classifier inside the intersection of the $l_\infty$- ball and the image domain $[0,1]^d$. Auto-PGD uses an automatic stepsize selection scheme and has been shown to outperform PGD. We use an adaptation to our setting (described in detail in Appendix A) with 500 steps and 5 restarts on 1000 points from each test set. Gradient masking poses a significant challenge, so we also perform a transfer attack on all models and on MNIST, we even use an additional attack (see Appendix A). We report the per-sample worst-case across attacks. Note that attacking these models on different out-distributions poses somewhat different challenges than classical adversarial attacks. Around the in-distribution models with good prediction performance are unlikely to be completely flat (and thus have zero gradient) in the whole region defined by an $l_\infty$-threat model. On the out-distribution, however, it is quite possible that all neurons in some layer return negative pre-activations which causes all gradients to be zero. Therefore the choice of initialization together with several restarts matters a lot as otherwise non-robust OOD detection models can easily appear to be robust. Moreover, the transfer attacks were necessary for some methods as otherwise the true robustness would have been significantly overestimated. Indeed even though we invested quite some effort into adaptive attacks which are specific for our robust OOD detection scenario, it might still be that the AAUC of some methods is overestimated. This again shows how important it is to get provable guarantees.

For the guaranteed AUC, we compute an upper bound on the confidence in the intersection of the $l_\infty$-ball with the image domain $[0,1]^d$ via IBP using (6) for the full test set. These worst case/guaranteed confidences for the out-distributions are then used for the AUC computation.

**Competitors.** We compare a normally trained model (Plain), the state-of-the-art OOD detection method Outlier Exposure (OE) [19], CEDA [16] and Adversarial Confidence Enhanced Training (ACET) [16], which we adjusted to the given task as described in the appendix. As CEDA performs very similar to OE, we omit it in the figures for better readability. The $\epsilon$-radii for the $l_\infty$-balls are the same for ACET and GOOD. So far the only method which could provide robustness guarantees for OOD detection is Certified Certain Uncertainty (CCU) with a data-dependent Mahalanobis-type $l_2$ threat model. We use their publicly available code to train a CCU model with our architecture and we evaluate their guarantees for our $l_\infty$ threat model. In Appendix B, we provide details and explain why their guarantees turn out to be vacuous in our setting.

## 4.2 Results

In Table 1 we present the results on all datasets.

Table 1: Accuracies as well as AUC, adversarial AUC (AAUC) and guaranteed AUC (GAUC) values for the MNIST, SVHN and CIFAR-10 in-distributions with respect to several unseen out-distributions. The GAUC of GOOD$_{100}$ on MNIST/SVHN resp. GOOD$_{80}$ on CIFAR-10 is better than the corresponding AAUC of OE and CEDA on almost all OOD datsets (except EMNIST). Thus GOOD is provably better than OE and CEDA w.r.t. worst-case OOD detection. GOOD achieves this without significant loss in accuracy. Especially on SVHN, GOOD$_{100}$ has very good accuracy and almost perfect provably worst-case OOD detection performance.

**IN: MNIST    $\epsilon = 0.3$**

| METHOD | ACC. | FASHIONMNIST | | | EMNIST LETTERS | | | CIFAR-10 | | | UNIFORM NOISE | | |
|---|---|---|---|---|---|---|---|---|---|---|---|---|---|
| | | AUC | AAUC | GAUC | AUC | AAUC | GAUC | AUC | AAUC | GAUC | AUC | AAUC | GAUC |
| PLAIN | 99.4 | 98.0 | 34.2 | 0.0 | 88.0 | 31.4 | 0.0 | 98.8 | 36.6 | 0.0 | 99.1 | 36.5 | 0.0 |
| CEDA | 99.4 | 99.9 | 82.1 | 0.0 | 92.6 | 52.8 | 0.0 | **100.0** | 95.1 | 0.0 | **100.0** | **100.0** | 0.0 |
| OE | 99.4 | 99.9 | 76.8 | 0.0 | 92.7 | 50.9 | 0.0 | **100.0** | 92.4 | 0.0 | **100.0** | **100.0** | 0.0 |
| ACET | 99.4 | **100.0** | **98.4** | 0.0 | 95.9 | 61.5 | 0.0 | **100.0** | 99.3 | 0.0 | **100.0** | **100.0** | 0.0 |
| CCU | **99.5** | **100.0** | 76.6 | 0.0 | 92.9 | 3.1 | 0.0 | **100.0** | 98.9 | 0.0 | **100.0** | **100.0** | 0.0 |
| GOOD$_{0}$ | 99.5 | 99.9 | 82.3 | 0.0 | 92.9 | 55.0 | 0.0 | **100.0** | 94.7 | 0.0 | **100.0** | **100.0** | 0.0 |
| GOOD$_{20}$ | 99.0 | 99.8 | 88.2 | 9.7 | 95.3 | 54.3 | 0.0 | **100.0** | 97.6 | 28.3 | **100.0** | **100.0** | 100.0 |
| GOOD$_{40}$ | 99.0 | 99.8 | 88.0 | 29.1 | 95.7 | 56.6 | 0.0 | **100.0** | 97.7 | 65.2 | **100.0** | **100.0** | 100.0 |
| GOOD$_{60}$ | 99.0 | 99.9 | 88.8 | 42.0 | 96.6 | 57.9 | 0.1 | **100.0** | 97.9 | 85.3 | **100.0** | **100.0** | 100.0 |
| GOOD$_{80}$ | 99.1 | 99.8 | 90.3 | 55.5 | 97.9 | **63.1** | 3.4 | **100.0** | 98.4 | 94.7 | **100.0** | **100.0** | 100.0 |
| GOOD$_{90}$ | 98.8 | 99.9 | 91.4 | 66.9 | 98.0 | 59.4 | 5.1 | **100.0** | 99.0 | 97.8 | **100.0** | **100.0** | 100.0 |
| GOOD$_{95}$ | 98.8 | 99.9 | 93.1 | 73.9 | 98.7 | 59.2 | **5.6** | **100.0** | 99.4 | 98.8 | **100.0** | **100.0** | 100.0 |
| GOOD$_{100}$ | 98.7 | **100.0** | 96.5 | **78.0** | **99.0** | 53.8 | 3.3 | **100.0** | **99.9** | 99.4 | **100.0** | **100.0** | 100.0 |

**IN: SVHN    $\epsilon = 0.03$**

| METHOD | ACC. | CIFAR-100 | | | CIFAR-10 | | | LSUN CLASSROOM | | | UNIFORM NOISE | | |
|---|---|---|---|---|---|---|---|---|---|---|---|---|---|
| | | AUC | AAUC | GAUC | AUC | AAUC | GAUC | AUC | AAUC | GAUC | AUC | AAUC | GAUC |
| PLAIN | 95.5 | 94.9 | 11.3 | 0.0 | 95.2 | 11.1 | 0.0 | 95.7 | 14.1 | 0.0 | 99.4 | 57.9 | 0.0 |
| CEDA | 95.3 | 99.9 | 63.9 | 0.0 | 99.9 | 68.7 | 0.0 | 99.9 | 80.7 | 0.0 | 99.9 | 99.3 | 0.0 |
| OE | 95.5 | **100.0** | 60.2 | 0.0 | **100.0** | 62.5 | 0.0 | **100.0** | 77.3 | 0.0 | **100.0** | 98.2 | 0.0 |
| ACET | 96.0 | **100.0** | **99.4** | 0.0 | **100.0** | **99.5** | 0.0 | **100.0** | **99.8** | 0.0 | 99.9 | 96.3 | 0.0 |
| CCU | 95.7 | **100.0** | 52.5 | 0.0 | **100.0** | 56.8 | 0.0 | **100.0** | 72.1 | 0.0 | **100.0** | **100.0** | 0.0 |
| GOOD$_{0}$ | **97.0** | **100.0** | 61.0 | 0.0 | **100.0** | 60.0 | 0.0 | **100.0** | 60.8 | 0.0 | **100.0** | 82.5 | 0.0 |
| GOOD$_{20}$ | 95.9 | 99.8 | 78.2 | 24.4 | 99.9 | 81.8 | 20.3 | 99.9 | 91.2 | 21.6 | 99.7 | 99.5 | 99.5 |
| GOOD$_{40}$ | 96.3 | 99.5 | 81.6 | 46.0 | 99.5 | 85.0 | 50.6 | 99.5 | 95.1 | 55.7 | 99.5 | 99.5 | 99.4 |
| GOOD$_{60}$ | 96.1 | 99.4 | 83.9 | 67.4 | 99.4 | 87.4 | 72.9 | 99.4 | 96.5 | 82.3 | 99.4 | 99.4 | 99.4 |
| GOOD$_{80}$ | 96.3 | **100.0** | 93.5 | 87.7 | **100.0** | 95.3 | 91.3 | **100.0** | 98.8 | 96.7 | **100.0** | **100.0** | 99.7 |
| GOOD$_{90}$ | 96.2 | 99.8 | 96.0 | 93.9 | 99.8 | 97.3 | 96.1 | 99.8 | 98.9 | 98.3 | 99.8 | 99.8 | **99.8** |
| GOOD$_{95}$ | 96.4 | 99.8 | 97.2 | 96.1 | 99.8 | 98.0 | 97.3 | 99.8 | 99.3 | **98.9** | 99.9 | 99.9 | **99.8** |
| GOOD$_{100}$ | 96.3 | 99.6 | 97.7 | **97.3** | 99.7 | 98.4 | **98.1** | 99.9 | 99.2 | **98.9** | **100.0** | 99.9 | **99.8** |

**IN: CIFAR-10    $\epsilon = 0.01$**

| METHOD | ACC. | CIFAR-100 | | | SVHN | | | LSUN CLASSROOM | | | UNIFORM NOISE | | |
|---|---|---|---|---|---|---|---|---|---|---|---|---|---|
| | | AUC | AAUC | GAUC | AUC | AAUC | GAUC | AUC | AAUC | GAUC | AUC | AAUC | GAUC |
| PLAIN | 90.1 | 84.3 | 13.0 | 0.0 | 87.7 | 10.6 | 0.0 | 88.9 | 13.6 | 0.0 | 90.8 | 56.4 | 0.0 |
| CEDA | 88.6 | 91.8 | 31.9 | 0.0 | **97.9** | 25.7 | 0.0 | 98.9 | 53.9 | 0.0 | 97.3 | 70.5 | 0.0 |
| OE | 90.7 | 92.4 | 11.0 | 0.0 | 97.6 | 3.7 | 0.0 | 98.9 | 20.0 | 0.0 | 98.7 | 75.7 | 0.0 |
| ACET | 89.3 | 90.7 | **74.5** | 0.0 | 96.6 | **88.0** | 0.0 | 98.3 | **91.2** | 0.0 | 99.7 | 98.9 | 0.0 |
| CCU | **91.6** | **93.0** | 23.3 | 0.0 | 97.1 | 14.8 | 0.0 | **99.3** | 38.2 | 0.0 | **100.0** | **100.0** | 0.0 |
| GOOD$_{0}$ | 89.8 | 92.9 | 22.5 | 0.0 | 97.0 | 12.8 | 0.0 | 98.3 | 48.4 | 0.0 | 96.3 | 95.6 | 0.0 |
| GOOD$_{20}$ | 88.5 | 90.3 | 32.4 | 11.8 | 95.9 | 28.3 | 15.8 | 98.2 | 48.2 | 3.4 | 99.4 | 97.6 | 87.5 |
| GOOD$_{40}$ | 89.5 | 89.6 | 38.2 | 24.8 | 95.4 | 38.0 | 24.9 | 96.0 | 62.0 | 27.4 | 92.1 | 89.9 | 89.8 |
| GOOD$_{60}$ | 90.2 | 88.6 | 42.6 | 34.9 | 95.6 | 44.4 | 39.0 | 97.0 | 67.6 | 49.1 | 91.8 | 91.3 | 91.2 |
| GOOD$_{80}$ | 90.1 | 85.6 | 48.2 | 42.3 | 94.0 | 41.4 | 38.0 | 93.3 | 66.9 | 55.2 | 95.8 | 95.4 | 95.3 |
| GOOD$_{90}$ | 90.2 | 81.7 | 51.5 | 49.6 | 91.4 | 48.7 | 46.9 | 90.2 | 63.5 | 57.7 | 89.3 | 87.7 | 87.7 |
| GOOD$_{95}$ | 90.4 | 80.3 | 52.0 | 50.8 | 90.2 | 44.4 | 43.3 | 88.3 | 62.6 | 60.3 | 96.6 | 95.9 | 95.8 |
| GOOD$_{100}$ | 90.1 | 70.0 | 54.7 | **54.2** | 75.5 | 58.9 | **56.9** | 75.2 | 61.5 | **61.0** | 99.5 | 99.2 | **99.0** |

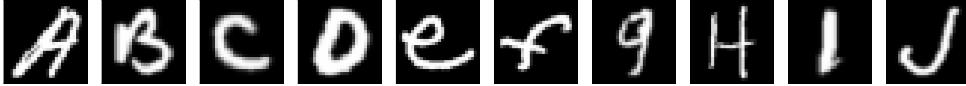

| Plain | 9: 63.6 | 8: 97.9 | 6: 87.2 | 0: 100.0 | 2: 60.0 | 4: 75.6 | 9: 100.0 | 4: 100.0 | 1: 100.0 | 5: 99.2 |
| OE | 9: 79.7 | 0: 26.8 | 0: 78.8 | 0: 100.0 | 0: 69.4 | 4: 80.9 | 9: 100.0 | 4: 100.0 | 1: 100.0 | 5: 98.7 |
| CCU | 4: 96.4 | 8: 31.8 | 6: 90.6 | 0: 100.0 | 2: 81.8 | 4: 86.8 | 9: 100.0 | 4: 100.0 | 1: 100.0 | 5: 96.2 |
| ACET | 4: 79.6 | 0: 38.4 | 0: 61.6 | 0: 99.7 | 2: 55.7 | 4: 94.4 | 9: 100.0 | 4: 98.4 | 1: 99.9 | 5: 79.4 |
| $GOOD_{100}$ | 8: 10.0 | 8: 11.4 | 0: 16.1 | 8: 10.0 | 8: 44.7 | 4: 24.1 | 9: 57.2 | 8: 10.0 | 1: 95.9 | 5: 12.1 |

Figure 2: Random samples from 10 letters in the out-distribution dataset EMNIST. The predictions and confidences of all methods trained on MNIST are shown on top. $GOOD_{100}$ is the only method which is **not** overconfident (e.g. "H") unless the letter is indistinguishable from a digit ("I").

**GOOD is provably better than OE/CEDA with regard to worst case OOD detection.** We note that for almost all OOD datasets GOOD achieves non-trivial GAUCs. Thus the guarantees generalize from the training out-distribution 80M to the test OOD datasets. For the easier in-distributions MNIST and SVHN, which are more clearly separated from the out-distribution, the overall best results are achieved for $GOOD_{100}$. For CIFAR-10, the clean AUCs of $GOOD_{100}$ are low even when compared to plain training. Arguably the best trade-off for CIFAR-10 is achieved by $GOOD_{80}$. Note that the guaranteed AUC (GAUC) of these models is always better than the adversarial AUC (AAUC) of OE/CEDA (except for EMNIST). Thus it is fair to say that the worst-case OOD detection performance of GOOD is provably better than that of OE/CEDA. As expected, ACET yields good AAUCs but has no guarantees. The failure of CCU regarding guarantees is discussed in Appendix B. It is notable that $GOOD_{100}$ has close to perfect guaranteed OOD detection performance for MNIST on CIFAR-10/uniform noise and for SVHN on **all** out-distribution datasets. In Appendix I we show that the guarantees of GOOD generalize surprisingly well to larger radii than seen during training.

**GOOD achieves certified OOD performance with almost no loss in accuracy.** While there is a small drop in clean accuracy for MNIST, on SVHN, with $96.3\%$ $GOOD_{100}$ has a better clean accuracy than all competing methods. On CIFAR-10, $GOOD_{80}$ achieves an accuracy of $90.1\%$ which is better than ACET and only slightly worse than CCU and OE. This is remarkable as we are not aware of any model with certified *adversarial robustness on the in-distribution* which gets even close to this range; e.g. IBP [13] achieves an accuracy of 85.2% on SVHN with $\epsilon = 0.01$ (we have 96.3%), on CIFAR-10 with $\epsilon = \frac{2}{255}$ they get 71.2% (we have 90.1%). Previous certified methods had even worse clean accuracy. Since a significant loss in prediction performance is usually not acceptable, certified methods have not yet had much practical impact. Thus we think it is an encouraging and interesting observation that properties different from adversarial robustness like worst-case out-of-distribution detection can be certified without suffering much in accuracy. In particular, it is quite surprising that certified methods can be trained effectively with aggressive data augmentation like AutoAugment.

**Trade-off between clean and guaranteed AUC via Quantile-GOOD.** As discussed above, for the CIFAR-10 experiments, our training out-distribution contains images from in-distribution classes. This seems to be the reason why $GOOD_{100}$ suffers from a significant drop in clean AUC, as the only way to ensure small loss $\mathcal{L}_{CUB}$, if in- and out-distribution can partially not be distinguished, is to reduce also the confidence on the in-distribution. This conflict is resolved via $GOOD_{80}$ and $GOOD_{90}$ which both have better clean AUCs. It is an interesting open question if similar trade-offs can also be useful for certified adversarial robustness.

**EMNIST: distinguishing letters from digits without ever having seen letters.** $GOOD_{100}$ achieves an excellent AUC of 99.0% for the letters of EMNIST which is, up to our knowledge, state-of-the-art. Indeed, an AUC of 100% should not be expected as even for humans some letters like i and l are indistinguishable from digits. This result is quite remarkable as $GOOD_{100}$ has never seen letters during training. Moreover, as the AUC just distinguishes the separation of in- and out-distribution based on the confidence, we provide the mean confidence on all datasets in the Appendix in Table 4 and in Figure 2 (see also Figure 3 in the Appendix) we show some samples from EMNIST together with their prediction/confidences for all models. $GOOD_{100}$ has a mean confidence of 98.4% on MNIST but only 27.1% on EMNIST in contrast to ACET with 75.0%, OE 87.9% and Plain 91.5%. This shows that while the AUC's of ACET and OE are good for EMNIST, these methods are still highly overconfident on EMNIST. Only $GOOD_{100}$ produces meaningful higher confidences on EMNIST, when the letter has clear features of the corresponding digit.

# 5 Conclusion

We propose GOOD, a novel training method to achieve guaranteed OOD detection in a worst-case setting. GOOD provably outperforms OE, the state-of-the-art in OOD detection, in worst case OOD detection and has state-of-the-art performance on EMNIST which is a particularly challenging out-distribution dataset. As the test accuracy of GOOD is comparable to the one of normal training, this shows that certified methods have the potential to be useful in practice even for more complex tasks. In future work it will be interesting to explore how close certified methods can get to state-of-the-art test performance.

## Broader Impact

In order to use machine learning in safety-critical systems it is required that the machine learning system correctly flags its uncertainty. As neural networks have been shown to be overconfident far away from the training data, this work aims at overcoming this issue by not only enforcing low confidence on out-distribution images but even guaranteeing low confidence in a neighborhood around it. As a neural network should not flag that it knows when it does not know, this paper contributes to a safer use of deep learning classifiers.

## Acknowledgements

The authors acknowledge support from the German Federal Ministry of Education and Research (BMBF) through the Tübingen AI Center (FKZ: 01IS18039A) and from the Deutsche Forschungsgemeinschaft (DFG, German Research Foundation) under Germany's Excellence Strategy (EXC number 2064/1, Project number 390727645). The authors thank the International Max Planck Research School for Intelligent Systems (IMPRS-IS) for supporting Alexander Meinke.

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
