[Supplementary Material]

# APPENDIX

## A  Adversarial attacks on OOD detection

It has been demonstrated [2, 36, 4, 7] that without strong countermeasures, DNNs are very susceptible to adversarial attacks changing the classification result. The goal of adversarial attacks in our setting is to fool the OOD detection which is based on the confidence in the prediction. Thus the attacker aims at maximizing the confidence in a neighborhood around a given out-distribution input $x$ so that the adversarially modified image will be wrongly assigned to the in-distribution. In this paper, we regard as threat model/neighborhood an $l_\infty$-ball of a given radius $\epsilon$, that is $\{z \in [0,1]^d \,|\, \|z - x\|_\infty \leq \epsilon\}$; note that in our case the disturbed inputs have to be valid images, hence the additional constraint $z \in [0,1]^d$.

For evaluation, we use Auto-PGD [8], which is a state-of-the-art implementation of PGD (projected gradient descent) using adaptive step sizes and random restarts. We use additionally backtracking. Since Auto-PGD has been designed for finding adversarial samples around the in-distribution, we change the objective of Auto-PGD to be the confidence of the classifier. We use Auto-PGD with 500 steps and 5 random restarts which is a quite strong attack. By default, the random initialization is drawn uniformly from the $\epsilon$-ball. However, we found that for MNIST the attack very often got stuck for our GOOD models, because a large random perturbation of size 0.3 would move the sample directly into a region of the input space where the model is completely flat and thus no gradients are available (in this sense adversarial attacks on OOD inputs are more difficult than usual adversarial attacks on the in-distribution). We instead use a modified version of the attack for MNIST which starts within short distance of the original point. Thus we use as initialization a random perturbation from $[-0.01, 0.01]^d$ (note that for our evaluation on CIFAR10, this choice coincides with the default settings).

Nevertheless, for MNIST most out-distribution points lie in regions where the predictions of our GOOD models are flat, i.e. the gradients are exactly zero. Because of this, Auto-PGD is unable to effectively explore the search space around those points. Thus, for MNIST we created an adaptive attack which partially circumvents these issues. First, we use an initialization scheme that mitigates lack of gradients by increasing the contrast as much as the threat model allows. All pixel values $x_i$ that lie above $1 - \epsilon$ get set to $x_i = 1$ and all values $x_i \leq 1 - \epsilon$ get set to $\max\{0, x_i - \epsilon\}$. In our experience these points are more likely to yield gradients, so we use them as initialization for a 200-step PGD attack with backtracking, adaptive step size selection and momentum of 0.9. Concretely, we use a step size of 0.1, and whenever a PGD step does not increase the confidence we backtrack and halve the step size. After every successful gradient step we multiply the step size by 1.1. Using backtracking and adaptive step size is necessary because otherwise one can easily step into regions where gradient information is no longer available.

Additionally, to further mitigate the problem of gradient-masking at initialization, for each model we use the final best points of all other models and use those as starting points for the same monotone PGD as described before. We use the sample-wise worst-case confidence to compute the final AAUC. Especially CEDA displays much higher apparent robustness if one omits the transfer attacks. Surprisingly, in this respect CEDA behaves very differently from OE, even though they pursue very similar objectives during training.

## B  A review of robust OOD detection

**ACET**   A method that was proposed in order to achieve adversarially robust low confidence on OOD data is Adversarial Confidence Enhancing Training (ACET) [16] which is based on adversarial training on the out-distribution. However, similar to adversarial training on the in-distribution, typically this does not lead to any guarantees, whereas our goal is to get guarantees on the confidences of worst-case out-distribution inputs. ACET has the following objective:

$$\frac{1}{N} \sum_{i=1}^{N} \mathcal{L}_{\text{CE}}(x_i^{\text{IN}}, y_i^{\text{IN}}) + \frac{\kappa}{M} \sum_{j=1}^{M} \max_{\|\hat{x} - x_j^{\text{OUT}}\|_\infty \leq \epsilon} \mathcal{L}_{\text{OUT}}(\hat{x}) \,. \tag{11}$$

They use $\mathcal{L}_{\text{OUT}} = \log \text{Conf}_f$ with low frequency noise as their training out-distribution. We found firstly that training an ACET model with 80M as out-distribution yields much better results than the

smoothed uniform noise used in [16] and secondly using the cross-entropy loss with respect to the uniform prediction instead of $\log \mathrm{Conf}_f$ also leads to improvements. For training ACET models, we employ a standard PGD attack with 40 steps of size $\frac{2\epsilon}{41}$ with initialization at the target input for maximizing the loss around $x_j^{\mathrm{OUT}}$. As usual for a $l_\infty$-attack, we use the sign of the gradient as direction and project onto the intersection of the image domain $[0,1]^d$ and the $l_\infty$-ball of radius $\epsilon$ around the target. Finally, the attack returns the image with the highest confidence found during the iterations. For the attack at training time we use no backtracking or adaptive stepsizes. ACET does not provide any guaranteed confidence bounds.

**CCU** Certified Certain Uncertainty (CCU) [28] gives low confidence guarantees around certain OOD data that is far away from the training dataset in a specific metric. Those bounds do hold on such far-away datasets, but do not generalize to inputs relatively close to the in distribution, like for example CIFAR-10 vs. CIFAR-100. Moreover, even in the regime where CCU yields meaningful guarantees, they are given in terms of a data-dependent Mahalanobis distance rather than the $l_\infty$-distance. However, due to norm equivalences, one can still extract $l_\infty$-guarantees from CCU and we evaluated the CCU guarantees as follows. We use the corollary 3.1 from [28] which states that for a CCU model that is written as

$$p(y|x) = \frac{p(y|x,i)p(x|i) + \frac{1}{K}p(x|o)}{p(x|i) + p(x|o)} \tag{12}$$

with $p(y|x,i)$ being the softmax output of a neural network and $p(x|i)$ and $p(x|o)$ Gaussian mixture models for in-and out-distribution, one can bound the confidence in a certain neighborhood around any point $x \in \mathbb{R}^d$ via

$$\max_{d_M(\hat{x},x) \leq R} p(y|x) \leq \frac{1}{K}\frac{1 + K\,b(x,R)}{1 + b(x,R)}. \tag{13}$$

Here $b : \mathbb{R}^d \times \mathbb{R}_+ \to \mathbb{R}_+$ is a positive function that increases monotonically in the radius $R$ and that depends on the parameters of the Gaussian mixture models (details in [28]). The metric $d_M : \mathbb{R}^d \times \mathbb{R}^d \to \mathbb{R}_+$ that they used for their CCU model is given as

$$d_M(x,y) = \left\| C^{-\frac{1}{2}}(x-y) \right\|, \tag{14}$$

where $C$ is a regularized version of the covariance matrix, calculated on the augmented in-distribution data. Note that this Mahalanobis metric is strongly equivalent to the metric induced by the $l_2$-norm and consequently to the metric induced by the $l_\infty$-norm. By computing the equivalence constants between these metrics we can extract the $l_\infty$-guarantees that are implicit in the CCU model. Geometrically speaking, we compute the size a an ellipsoid (its shape determined by the eigenvalues of $C$) that is large enough to fit a cube inside it with a radius given by our threat model $r = 0.3$ or $r = 0.01$, respectively. Via norm equivalences one has

$$d_M(x,y) \leq \sqrt{\lambda_1}d_2(x,y) \leq \sqrt{d\lambda_1}d_\infty(x,y) \leq \sqrt{d\lambda_1}r, \tag{15}$$

where $\lambda_1$ is the largest eigenvalue of $C$. This means that the confidence upper bounds from (13) on a Mahalanobis-ball of radius $R = (d\lambda)^{\frac{1}{2}}r$ automatically apply to an $l_\infty$-ball of radius $r$. However, the covariance matrix $C$ is highly ill-conditioned, which means that $\lambda_1$ is fairly high. On top of that, in high dimensions $\sqrt{d}$ is big as well so that in practice the required radius $R$ becomes too large for CCU to certify meaningful guarantees. Even on uniform noise, the upper bounds were larger than the highest confidence on the in-distribution test set, with the consequence that there are no lower-bounds on the AAUC. However, we want to stress that at least for uniform noise the lack of guarantees of CCU is due to the incompatability of the threat models used in our paper and [28].

Another type of guarantee that certifies a detection rate for OOD samples by applying probably approximately correct (PAC) learning considerations has been proposed in [26]. Their problem setting and nature of guarantees are not directly comparable to ours, since their guarantees handle behaviour on whole distributions while our guarantees are given for individual datapoints.

## C AUC and Conservative AUC

As a measure for the separation of in- vs. out-distribution data we use the Area Under the Receiver Operating Characteristic curve (AUROC or AUC) using the confidence of the classifier as the feature.

The AUC is equal to the empirical probability of a random in-sample to be assigned a higher confidence than a random out-sample, plus one half times the probability of the confidences being equal. Thus, the standard way (as e.g. implemented in scikit-learn [32]) to calculate the AUC from given confidence values on sets of in- and out-distribution samples $S_{in}$ and $S_{out}$ is

$$\text{AUC}(f, S_{in}, S_{out}) = \frac{1}{|S_{in}||S_{out}|} \Bigg( |\{x_{in} \in S_{in}, x_{out} \in S_{out} \mid \text{Conf}_f(x_{in}) > \text{Conf}_f(x_{out})\}|$$
$$+ \frac{1}{2} |\{x_{in} \in S_{in}, x_{out} \in S_{out} \mid \text{Conf}_f(x_{in}) = \text{Conf}_f(x_{out})\}| \Bigg) , \tag{16}$$

where for a set $S$, $|S|$ indicates the number of its elements. The half-weighted equality term gives this definition certain symmetry properties. However, it assigns a positive score to some completely uninformed functions $f$. For example, a constant uniform classifier with $p_k(x) = \frac{1}{K}$ receives an AUC value of 50%. Similarly, a classifier that assigns 100% confidence to most in-distribution inputs would have positive AUC and even GAUC statistics, even if it fails to have confidence below 100% on any OOD inputs. In order to regard only example pairs where the distributions are positively distinguished, we define the **Conservative AUC** (cAUC) by dropping the equality term:

$$\text{cAUC}(f, S_{in}, S_{out}) := \frac{1}{|S_{in}||S_{out}|} |\{x_{in} \in S_{in}, x_{out} \in S_{out} \mid \text{Conf}_f(x_{in}) > \text{Conf}_f(x_{out})\}| . \tag{17}$$

While in general $\text{cAUC}(f, S_{in}, S_{out}) \leq \text{AUC}(f, S_{in}, S_{out})$, the confidences of all models presented in the paper are differentiated enough so that for all shown numbers actually cAUC = AUC. However, we have experienced that one can have models where the confidences (uniform or one-hot predictions) cannot be distinguished due to limited numerical precision. In these cases the normal AUC definition would indicate a certain discrimination where it is actually impossible to discriminate the confidences.

## D    Experimental details

The layer compositions of the architectures used for all GOOD and baseline models are laid out in Table 2. No normalization of inputs or activations is used. Weight decay ($l_2$) is set to 0.05 for MNIST and 0.005 for SVHN and CIFAR-10. For all runs, we use a batch size of 128 samples from both the in- and the out-distribution (where applicable). At `https://gitlab.com/Bitterwolf/GOOD` you can find the exact implementation.

Table 2: Model architectures used for MNIST (L), SVHN (XL) and CIFAR-10 (XL) experiments. Each convolutional and non-final affine layer is followed by a ReLU activation. All convolutions use a kernel size of 3, a padding of 1, and stride of 1, except for the third convolution which has stride=2.

| L | XL |
|---|---|
| CONV2D(64) | CONV2D(128) |
| CONV2D(64) | CONV2D(128) |
| CONV2D(128)$_{s=2}$ | CONV2D(256)$_{s=2}$ |
| CONV2D(128) | CONV2D(256) |
| CONV2D(128) | CONV2D(256) |
| LINEAR(512) | LINEAR(512) |
| LINEAR(10) | LINEAR(512) |
| | LINEAR(10) |

For the MNIST experiments, we use as optimizer SGD with 0.9 Nesterov momentum, with an initial learning rate of $\frac{0.005}{128}$ that is divided by 5 after 50, 100, 200, 300 and 350 epochs, with a total number of 420 training epochs. For the GOOD, CEDA and OE runs, the first two epochs only use in-distribution $\mathcal{L}_{\text{CE}}$; over the next 100 epochs, the value of $\kappa$ is ramped up linearly from zero to its final value of 0.3 for GOOD/OE and 1.0 for CEDA, where it stays for the remaining 318 epochs. The $\epsilon$ value in the $\mathcal{L}_{\text{CUB}}$ loss for GOOD is also increased linearly, starting at epoch 10 and reaching its

final value of $0.3$ on epoch 130. CCU is trained using the publicly available code from [28], where we modify the architecture, learning rate schedule and data augmentation to be the same as OE. The initial learning rate for the Gaussian mixture models is $1e - 5/\text{batchsize}$ and gets dropped at the same epochs as the neural network learning rate. Our more aggressive data augmentation implies that our underlying Mahalanobis metric is not the same as they used in [28]. The ACET model for MNIST is warmed up with two epochs on the in-distribution only, then four with $\kappa = 1.0$ and $\epsilon = 0$, and the full ACET loss with $\kappa = 1.0$ and $\epsilon = 0.3$ for the remaining epochs. The reason why we chose a smaller $\kappa$ of $0.3$ for the MNIST GOOD runs is that considering the large $\epsilon$ for which guarantees are enforced, training with higher $\kappa$ values makes training unstable without improving any validation results.

For the SVHN and CIFAR-10 baseline models, we used the ADAM optimizer [20] with initial learning rate $\frac{0.01}{128}$ for SVHN and $\frac{0.1}{128}$ for CIFAR-10 that was divided by 5 after 30 and 100 epochs, with a total number of 420 training epochs. For OE, $\kappa$ is increased linearly from zero to one between epochs 60 and 360. The same holds for CCU which again uses the same hyperparameters as OE. Again, ACET is warmed up with two in-distribution-only and four OE epochs. Then it is trained with $\kappa = 1.0$ and $\epsilon = 0.03/0.01$ (SVHN/CIFAR-10), with a shorter training time of 100 epochs (the same number as used in [16]).

In line with the experiences reported in [13] and [41], for GOOD training on SVHN and CIFAR-10 longer training schedules with slower ramping up of the $\mathcal{L}_{\text{CUB}}$ loss are necessary, as adding the out-distribution loss defined in Equation (8) to the training objective at once will overwhelm the in-distribution cross-entropy loss and cause the model to collapse to uniform predictions for all inputs, without recovery. In order to reduce warm-up time, we use a pre-trained CEDA model for initialization and train for 900 epochs. The learning rate is $10^{-4}$ in the beginning and is divided by 5 after epochs 450, 750 and 850. Due to the pre-training, we begin training with a small $\kappa$ and already start with non-zero $\epsilon$ after epoch 4. Then, $\epsilon$ is increased linearly to its final value of $0.03$ for SVHN and $0.01$ for CIFAR-10, which is reached at epoch 204. Simultaneously, $\kappa$ is increased linearly with a virtual starting point at epoch -2 to its final value of $1.0$ at epoch 298.

Due to the tendency of IBP based training towards instabilities, the selection of hyper-parameters was based on finding settings where training is reliably stable while guaranteed bounds over meaningful $\epsilon$-radii are possible.

For the accuracy, AUC and GAUC evaluations in Table 1 the test splits of each (non-noise) dataset were used, with the following numbers of samples: 10,000 for MNIST, FashionMNIST, CIFAR-10, CIFAR-100 and Uniform Noise; 20,800 for EMNIST Letters; 26,032 for SVHN; 300 for LSUN Classroom. Due to the computational cost of the employed attacks, the AAUC values are based on subsets of 1000 samples for each dataset.

All experiments were run on Nvidia Tesla P100 and V100 GPUs, with GPU memory requirement below 16GB.

## E   Depiction of GOOD Quantile-loss

In Quantile-GOOD training, the out-distribution part of each batch is split up into "harder" and "easier" parts, since trying to enforce low confidence guarantees on out-distribution inputs that are very close to the in-distribution leads to low confidences in general, even on the in-distribution. In Table 3, we show example batches of $\text{GOOD}_{60}$ models with MNIST, SVHN and CIFAR-10 as in-distribution near the end of training (from epochs 410, 890 and 890, respectively). Even though the actual CIFAR images were filtered out, some images containing objects from CIFAR-classes are still present. For the CIFAR-10 model, such samples (among others) get sorted above the quantile. For MNIST, lower brightness images appear to be more difficult, while for SVHN images with fewer objects seem to be comparably hardest to distinguish from the house numbers of the in-distribution.

## F   Confidences on EMNIST

Figure 3 shows samples of the letters "k" through "z" together with the predictions and confidences of the $\text{GOOD}_{100}$ MNIST model and four baseline models, complementing Figure 2. We see that $\text{GOOD}_{100}$ produces low confidences for most letters when they show no digit-specific features. Interestingly it even rejects some letters that could easily be mistaken for digits by humans ("o"). The

mean confidence values of the same selection of MNIST models for each letter of the alphabet for EMNIST are plotted in Figure 4. We observe that the mean confidence often aligns with the intuitive likeness of a letter with some digit: $GOOD_{100}$ has the highest mean confidence on the letter inputs "i" and "l", which in many cases do look like the digit "1". Again, the confidence of $GOOD_{100}$ on the letter "o", which even humans often cannot distinguish from a digit "0", is generally low. On the other hand, "y" receives a surprisingly high confidence, compared to other letters, so we conclude that $GOOD_{100}$ uses different features than humans in order to achieve its impressive performance on EMNIST.

|  | K | L | m | n | o | p | g | r |
|---|---|---|---|---|---|---|---|---|
| Plain | 6: 82.1 | 6: 100.0 | 4: 99.9 | 7: 99.8 | 0: 100.0 | 4: 89.7 | 8: 90.8 | 8: 97.4 |
| OE | 6: 67.8 | 6: 100.0 | 4: 98.8 | 7: 99.5 | 0: 100.0 | 1: 65.9 | 8: 97.5 | 8: 96.2 |
| CCU | 8: 52.7 | 6: 100.0 | 4: 89.0 | 7: 99.0 | 0: 99.9 | 1: 78.7 | 8: 80.2 | 8: 78.5 |
| ACET | 6: 37.6 | 6: 99.7 | 4: 48.6 | 7: 58.8 | 0: 98.8 | 1: 48.0 | 8: 87.5 | 8: 69.0 |
| $GOOD_{100}$ | 8: 10.0 | 6: 10.9 | 4: 21.2 | 8: 10.0 | 8: 10.0 | 1: 20.4 | 8: 28.1 | 8: 28.6 |

|  | S | t | U | V | w | X | Y | Z |
|---|---|---|---|---|---|---|---|---|
| Plain | 5: 100.0 | 4: 97.6 | 0: 90.0 | 4: 99.8 | 6: 74.8 | 4: 76.9 | 4: 99.8 | 2: 100.0 |
| OE | 5: 100.0 | 4: 85.1 | 0: 79.5 | 4: 99.5 | 4: 68.0 | 4: 65.2 | 4: 100.0 | 2: 100.0 |
| CCU | 5: 100.0 | 4: 98.8 | 0: 99.7 | 4: 92.7 | 4: 72.6 | 4: 66.0 | 4: 100.0 | 2: 99.7 |
| ACET | 5: 99.9 | 4: 82.8 | 0: 87.1 | 4: 96.9 | 0: 60.8 | 6: 50.9 | 4: 99.8 | 2: 81.8 |
| $GOOD_{100}$ | 5: 10.1 | 4: 50.2 | 0: 11.2 | 4: 12.1 | 8: 10.0 | 8: 10.0 | 4: 74.8 | 8: 10.0 |

Figure 3: Continuation of Figure 2. Random samples from the remaining letters in the out-distribution dataset EMNIST. The predictions and confidences of different methods trained on MNIST are shown on top.

Figure 4: Mean confidence of different models across the classes of EMNIST-Letters. $GOOD_{100}$ only has high mean confidence on letters that can easily be mistaken for digits.

# G Distributions of confidences and confidence upper bounds

Table 4 shows the mean confidences of all models on the in-distribution as well as the mean confidences and the mean guaranteed upper bounds on the worst-case confidences on the evaluated out-distributions. As discussed, $GOOD_{100}$ training can reduce the confidence on the in-distribution, with a particularly strong effect for CIFAR-10. By adjusting the loss quantile, this effect can be significantly reduced while maintaining non-trivial guarantees.

The histograms of mean confidences on the in-distribution and mean guaranteed upper bounds on the worst-case confidences on the samples from the evaluated out-distribution test sets for seven models are shown in Tables 5 (MNIST), 6 (SVHN) and 7 (CIFAR-10). A higher GOOD loss quantile generally shifts the distribution of the upper bounds on the worst-case confidence towards smaller values, but in some cases, especially for $GOOD_{100}$ on CIFAR-10, strongly lowers confidences in in-distribution predictions as well.

Table 3: Exemplary batch of out-distribution 80M Tiny Images (after augmentation) towards the end of training of $GOOD_{60}$ models. **Top:** The 52 Images with highest confidence upper bound. On these, loss is based on standard output. **Bottom:** The remaining 76 Images with lowest confidence upper bound. Here, loss is based on upper bounds within the $\epsilon$-ball.

IN: MNIST          IN: SVHN          IN: CIFAR-10

Table 4: Mean confidence on the in-distribution and mean confidence / mean upper bounds on the confidence within the $l_\infty$-balls of radius $\epsilon$ on the evaluated out-distribution datasets.

| | | IN: MNIST | $\epsilon = 0.3$ | | |
|---|---|---|---|---|---|
| METHOD | MNIST | FASHIONMNIST | EMNIST LETTERS | CIFAR-10 | UNIFORM NOISE |
| PLAIN | 99.7 | 79.2 / 100.0 | 91.5 / 100.0 | 77.2 / 100.0 | 79.6 / 100.0 |
| CEDA | 99.7 | 22.0 / 100.0 | 88.3 / 100.0 | 10.0 / 100.0 | 10.0 / 100.0 |
| OE | 99.7 | 25.4 / 100.0 | 87.9 / 100.0 | 10.1 / 100.0 | 10.0 / 100.0 |
| ACET | 99.6 | 12.3 / 100.0 | 75.0 / 100.0 | 10.0 / 100.0 | 10.0 / 100.0 |
| CCU | 99.7 | 17.5 / 100.0 | 87.4 / 100.0 | 10.0 / 100.0 | 10.0 / 100.0 |
| $GOOD_0$ | 99.7 | 20.6 / 100.0 | 87.9 / 100.0 | 10.0 / 100.0 | 10.0 / 100.0 |
| $GOOD_{20}$ | 99.5 | 19.4 / 93.0 | 70.2 / 100.0 | 10.0 / 76.6 | 10.0 / 10.0 |
| $GOOD_{40}$ | 99.3 | 17.7 / 76.8 | 58.2 / 100.0 | 10.0 / 43.7 | 10.0 / 10.0 |
| $GOOD_{60}$ | 99.2 | 15.8 / 66.0 | 51.7 / 100.0 | 10.0 / 24.8 | 10.0 / 10.0 |
| $GOOD_{80}$ | 99.0 | 16.3 / 55.1 | 40.7 / 98.6 | 10.0 / 15.7 | 10.0 / 10.0 |
| $GOOD_{90}$ | 98.8 | 14.2 / 47.5 | 38.3 / 98.2 | 10.0 / 12.7 | 10.0 / 10.0 |
| $GOOD_{95}$ | 98.7 | 13.1 / 42.6 | 32.2 / 98.2 | 10.0 / 11.6 | 10.0 / 10.0 |
| $GOOD_{100}$ | 98.4 | 10.8 / 40.8 | 27.1 / 99.2 | 10.0 / 11.0 | 10.0 / 10.0 |

| | | IN: SVHN | $\epsilon = 0.03$ | | |
|---|---|---|---|---|---|
| METHOD | SVHN | CIFAR-100 | CIFAR-10 | LSUN CLASSROOM | UNIFORM NOISE |
| PLAIN | 97.7 | 70.8 / 100.0 | 70.5 / 100.0 | 66.8 / 100.0 | 40.5 / 100.0 |
| CEDA | 97.1 | 10.2 / 100.0 | 10.1 / 100.0 | 10.0 / 100.0 | 10.0 / 100.0 |
| OE | 97.0 | 10.7 / 100.0 | 10.5 / 100.0 | 10.3 / 100.0 | 10.2 / 100.0 |
| ACET | 93.5 | 10.2 / 100.0 | 10.1 / 100.0 | 10.1 / 100.0 | 10.5 / 100.0 |
| CCU | 97.2 | 10.8 / 100.0 | 10.6 / 100.0 | 10.4 / 100.0 | 10.0 / 100.0 |
| $GOOD_0$ | 98.7 | 10.0 / 100.0 | 10.0 / 100.0 | 10.0 / 100.0 | 10.0 / 100.0 |
| $GOOD_{20}$ | 97.6 | 10.1 / 78.1 | 10.1 / 81.9 | 10.0 / 80.7 | 10.0 / 10.0 |
| $GOOD_{40}$ | 97.6 | 10.1 / 61.4 | 10.1 / 57.4 | 10.0 / 54.0 | 10.0 / 10.2 |
| $GOOD_{60}$ | 97.4 | 10.1 / 44.2 | 10.1 / 39.8 | 10.0 / 33.7 | 10.0 / 10.1 |
| $GOOD_{80}$ | 96.1 | 10.1 / 28.1 | 10.1 / 23.6 | 10.0 / 17.4 | 10.0 / 10.2 |
| $GOOD_{90}$ | 94.7 | 10.1 / 20.9 | 10.0 / 17.7 | 10.0 / 14.0 | 10.0 / 10.0 |
| $GOOD_{95}$ | 93.4 | 10.2 / 18.2 | 10.1 / 15.7 | 10.1 / 12.6 | 10.0 / 10.0 |
| $GOOD_{100}$ | 91.5 | 10.7 / 16.7 | 10.3 / 14.5 | 10.1 / 12.1 | 10.0 / 10.1 |

| | | IN: CIFAR-10 | $\epsilon = 0.01$ | | |
|---|---|---|---|---|---|
| METHOD | CIFAR-10 | CIFAR-100 | SVHN | LSUN CLASSROOM | UNIFORM NOISE |
| PLAIN | 95.1 | 79.0 / 100.0 | 75.8 / 100.0 | 73.9 / 100.0 | 73.2 / 100.0 |
| CEDA | 87.0 | 29.0 / 100.0 | 12.1 / 100.0 | 10.5 / 100.0 | 11.9 / 100.0 |
| OE | 85.1 | 31.6 / 100.0 | 19.1 / 100.0 | 14.6 / 100.0 | 15.6 / 100.0 |
| ACET | 71.8 | 25.3 / 100.0 | 16.7 / 100.0 | 13.7 / 100.0 | 11.2 / 100.0 |
| CCU | 89.4 | 32.5 / 100.0 | 20.5 / 100.0 | 12.6 / 100.0 | 10.0 / 100.0 |
| $GOOD_0$ | 81.0 | 18.9 / 100.0 | 10.8 / 100.0 | 10.1 / 100.0 | 10.0 / 100.0 |
| $GOOD_{20}$ | 78.9 | 23.8 / 91.4 | 13.0 / 87.8 | 10.7 / 97.9 | 10.1 / 22.7 |
| $GOOD_{40}$ | 77.1 | 21.4 / 84.7 | 11.2 / 85.4 | 10.7 / 89.5 | 11.7 / 12.4 |
| $GOOD_{60}$ | 71.7 | 21.7 / 75.4 | 11.5 / 72.0 | 10.5 / 67.3 | 13.2 / 13.4 |
| $GOOD_{80}$ | 64.1 | 23.1 / 64.4 | 13.3 / 67.5 | 13.5 / 51.8 | 12.0 / 12.3 |
| $GOOD_{90}$ | 55.6 | 24.2 / 54.8 | 15.4 / 56.2 | 16.1 / 44.9 | 17.2 / 18.1 |
| $GOOD_{95}$ | 53.1 | 25.8 / 52.0 | 16.9 / 57.2 | 18.1 / 43.6 | 12.6 / 12.6 |
| $GOOD_{100}$ | 49.6 | 34.7 / 46.0 | 30.4 / 44.0 | 30.6 / 41.5 | 11.6 / 12.0 |

Table 5: Histograms of the confidences on the **MNIST** in-distribution and guaranteed upper bounds on the confidences on OOD datasets within the $l_\infty$-ball of radius 0.3. Each histogram uses 50 bins between 0.1 and 1.0. For better readability, the scale is zoomed in by a factor 10 for numbers below one fifth of the total number of datapoints of the shown datasets. The vertical dotted line shows the mean value of the histogram's data.

| MODEL | MNIST | FASHIONMNIST GUB | EMNIST LETTERS GUB | CIFAR-10 GUB | UNIFORM GUB |
|---|---|---|---|---|---|
| PLAIN | | | | | |
| OE | | | | | |
| ACET | | | | | |
| GOOD$_{40}$ | | | | | |
| GOOD$_{80}$ | | | | | |
| GOOD$_{90}$ | | | | | |
| GOOD$_{100}$ | | | | | |

Table 6: Histograms of the confidences on the **SVHN** in-distribution and guaranteed upper bounds on the confidences on OOD datasets within the $l_\infty$-ball of radius 0.03. Each histogram uses 50 bins between 0.1 and 1.0. For better readability, the scale is zoomed in by a factor 10 for numbers below one fifth of the total number of datapoints of the shown datasets. The vertical dotted line shows the mean value of the histogram's data.

| MODEL | SVHN | CIFAR-100 GUB | CIFAR-10 GUB | LSUN CLASSROOM GUB | UNIFORM GUB |
|---|---|---|---|---|---|
| PLAIN | | | | | |
| OE | | | | | |
| ACET | | | | | |
| GOOD$_{40}$ | | | | | |
| GOOD$_{80}$ | | | | | |
| GOOD$_{90}$ | | | | | |
| GOOD$_{100}$ | | | | | |

Table 7: Histograms of the confidences on the **CIFAR-10** in-distribution and guaranteed upper bounds on the confidences on OOD datasets within the $l_\infty$-ball of radius 0.01. Each histogram uses 50 bins between 0.1 and 1.0. For better readability, the scale is zoomed in by a factor 10 for numbers below one fifth of the total number of datapoints of the shown datasets. The vertical dotted line shows the mean value of the histogram's data.

| MODEL | CIFAR-10 | CIFAR-100 GUB | SVHN GUB | LSUN CLASSROOM GUB | UNIFORM GUB |
|---|---|---|---|---|---|

# H   Evaluation on additional datasets

Table 8: A continuation of Table 1 for additional out-distributions. As in Table 1 the guaranteed AUCs (GAUC) of the highlighted GOOD models are in general better than the adversarial (AAUC) of OE (with the exception of Omniglot for MNIST).

| | | IN: MNIST | | $\epsilon = 0.3$ | | | | | |
|---|---|---|---|---|---|---|---|---|---|
| | | **80M TINY IMAGES** | | | **OMNIGLOT** | | | **NOTMNIST** | | |
| METHOD | ACC. | AUC | AAUC | GAUC | AUC | AAUC | GAUC | AUC | AAUC | GAUC |
| PLAIN | 99.4 | 98.7 | 36.9 | 0.0 | 97.9 | 38.6 | 0.0 | 91.9 | 38.8 | 0.0 |
| CEDA | 99.4 | **100.0** | 94.3 | 0.0 | 98.5 | 53.1 | 0.0 | 99.9 | 97.8 | 0.0 |
| OE | 99.4 | **100.0** | 91.5 | 0.0 | 98.5 | 51.0 | 0.0 | 99.9 | 96.8 | 0.0 |
| ACET | 99.4 | **100.0** | 99.2 | 0.0 | **99.5** | **76.5** | 0.0 | **100.0** | 99.5 | 0.0 |
| CCU | 99.5 | **100.0** | 75.0 | 0.0 | 98.1 | 3.4 | 0.0 | **100.0** | 99.6 | 0.0 |
| $GOOD_0$ | 99.5 | **100.0** | 93.8 | 0.0 | 98.6 | 55.7 | 0.0 | 99.9 | 97.7 | 0.0 |
| $GOOD_{20}$ | 99.0 | **100.0** | 97.1 | 32.7 | 97.0 | 42.4 | 0.0 | **100.0** | 99.6 | 19.3 |
| $GOOD_{40}$ | 99.0 | **100.0** | 97.2 | 59.5 | 96.9 | 36.8 | 0.0 | **100.0** | 99.7 | 44.7 |
| $GOOD_{60}$ | 99.0 | **100.0** | 97.3 | 77.8 | 96.3 | 31.3 | 0.0 | **100.0** | 99.8 | 76.2 |
| $GOOD_{80}$ | 99.1 | **100.0** | 97.8 | 89.4 | 96.9 | 34.2 | 1.2 | **100.0** | 99.9 | 96.7 |
| $GOOD_{90}$ | 98.8 | **100.0** | 98.7 | 94.2 | 97.8 | 40.5 | 2.2 | **100.0** | 99.9 | 99.2 |
| $GOOD_{95}$ | 98.8 | **100.0** | 99.2 | 96.1 | 97.8 | 42.2 | **2.4** | **100.0** | **100.0** | **99.5** |
| $GOOD_{100}$ | 98.7 | **100.0** | **99.5** | **97.7** | 98.6 | 50.7 | 1.8 | **100.0** | 99.9 | 99.3 |

| | | IN: SVHN | | $\epsilon = 0.03$ | | | | | |
|---|---|---|---|---|---|---|---|---|---|
| | | **80M TINY IMAGES** | | | **IMAGENET-** | | | **SMOOTH NOISE** | | |
| METHOD | ACC. | AUC | AAUC | GAUC | AUC | AAUC | GAUC | AUC | AAUC | GAUC |
| PLAIN | 95.5 | 94.8 | 11.9 | 0.0 | 95.5 | 13.4 | 0.0 | 96.0 | 5.6 | 0.0 |
| CEDA | 95.3 | 99.9 | 64.4 | 0.0 | 99.9 | 75.3 | 0.0 | 96.8 | 5.9 | 0.0 |
| OE | 95.5 | **100.0** | 61.8 | 0.0 | **100.0** | 72.5 | 0.0 | 97.0 | 8.0 | 0.0 |
| ACET | 96.0 | **100.0** | **99.3** | 0.0 | **100.0** | **99.6** | 0.0 | 99.9 | **83.5** | 0.0 |
| CCU | 95.7 | **100.0** | 48.8 | 0.0 | **100.0** | 97.2 | 0.0 | **100.0** | 5.7 | 0.0 |
| $GOOD_0$ | 97.0 | **100.0** | 57.9 | 0.0 | **100.0** | 68.3 | 0.0 | 97.8 | 25.0 | 0.0 |
| $GOOD_{20}$ | 95.9 | 99.8 | 78.3 | 19.4 | 99.8 | 88.9 | 34.0 | 97.4 | 22.0 | 0.0 |
| $GOOD_{40}$ | 96.3 | 99.5 | 81.0 | 44.4 | 99.5 | 90.1 | 62.6 | 97.1 | 21.0 | 0.0 |
| $GOOD_{60}$ | 96.1 | 99.4 | 83.4 | 64.5 | 99.4 | 92.6 | 82.8 | 97.0 | 18.0 | 0.0 |
| $GOOD_{80}$ | 96.3 | **100.0** | 93.1 | 86.3 | **100.0** | 97.4 | 95.6 | 96.8 | 29.1 | 3.9 |
| $GOOD_{90}$ | 96.2 | 99.8 | 95.2 | 93.0 | 99.8 | 98.4 | 97.8 | 96.7 | 40.6 | 20.6 |
| $GOOD_{95}$ | 96.4 | 99.7 | 96.4 | 95.2 | 99.8 | 98.8 | 98.4 | 96.8 | 59.1 | 46.8 |
| $GOOD_{100}$ | 96.3 | 99.6 | 97.2 | **96.8** | 99.8 | 99.1 | **98.9** | 96.7 | 77.5 | **73.5** |

| | | IN: CIFAR-10 | | $\epsilon = 0.01$ | | | | | |
|---|---|---|---|---|---|---|---|---|---|
| | | **80M TINY IMAGES** | | | **IMAGENET-** | | | **SMOOTH NOISE** | | |
| METHOD | ACC. | AUC | AAUC | GAUC | AUC | AAUC | GAUC | AUC | AAUC | GAUC |
| PLAIN | 90.1 | 85.6 | 15.5 | 0.0 | 83.5 | 15.5 | 0.0 | 90.5 | 18.8 | 0.0 |
| CEDA | 88.6 | 97.2 | 49.6 | 0.0 | 90.1 | 32.6 | 0.0 | 98.9 | 37.3 | 0.0 |
| OE | 90.7 | **97.3** | 20.5 | 0.0 | 90.3 | 12.1 | 0.0 | 99.5 | 11.3 | 0.0 |
| ACET | 89.3 | 96.7 | **88.8** | 0.0 | 89.5 | **74.7** | 0.0 | **99.9** | **98.8** | 0.0 |
| CCU | 91.6 | 96.8 | 33.7 | 0.0 | **92.0** | 30.0 | 0.0 | 99.5 | 38.0 | 0.0 |
| $GOOD_0$ | 89.8 | 96.9 | 42.7 | 0.0 | 91.0 | 19.8 | 0.0 | 96.9 | 30.0 | 0.0 |
| $GOOD_{20}$ | 88.5 | 96.6 | 48.5 | 16.3 | 88.8 | 30.5 | 6.9 | 96.5 | 64.5 | 17.8 |
| $GOOD_{40}$ | 89.5 | 94.8 | 56.8 | 36.4 | 88.0 | 39.3 | 24.6 | 96.4 | 86.4 | 27.5 |
| $GOOD_{60}$ | 90.2 | 95.2 | 60.7 | 48.7 | 87.4 | 46.1 | 36.7 | 97.5 | 81.4 | 47.8 |
| $GOOD_{80}$ | 90.1 | 93.1 | 62.8 | 55.9 | 84.0 | 50.0 | 42.3 | 95.1 | 74.1 | 59.4 |
| $GOOD_{90}$ | 90.2 | 90.6 | 63.4 | 60.8 | 79.6 | 53.0 | 49.1 | 98.9 | 72.8 | 62.3 |
| $GOOD_{95}$ | 90.4 | 88.9 | 63.4 | 62.0 | 77.6 | 54.3 | 50.3 | 92.0 | 61.8 | 59.4 |
| $GOOD_{100}$ | 90.1 | 78.7 | 66.7 | **66.3** | 69.0 | 56.9 | **53.9** | 82.2 | 67.9 | **66.8** |

Extending the evaluation results presented in Table 1, we provide AUC, AAUC and GAUC values for additional out-distribution datasets in Table 8. These datasets are:

- 80M Tiny Images, the out-distribution that was used during training. While it is the same *distribution* as seen during training, the test set consists of 1,000 samples that are not part of the training set.

- Omniglot (Lake, B. M., Salakhutdinov, R., and Tenenbaum, J. B. (2015). Human-level concept learning through probabilistic program induction. Science, 350(6266), 1332-1338.) is a dataset of hand drawn characters. We use the evaluation split consisting of 13180 characters from 20 different alphabets.

- notMNIST is a dataset of the letters A to J taken from different publicly available fonts. The dataset was retrieved from `https://yaroslavvb.blogspot.com/2011/09/notmnist-dataset.html`. We evaluate on the hand cleaned subset of 18724 images,

- ImageNet- [16], which is a subset of ImageNet [10] without images labelled as classes equal or semantically similar to CIFAR-10 classes.

- Smooth Noise is generated as described by [16]. First, a uniform noise image is generated. Then, a Gaussian filter with $\sigma$ drawn uniformly at random between 1.0 and 2.5 is applied. Finally, the image is re-scaled such that the minimal pixel value is 0.0 and the maximal one is 1.0. We evaluate AUC and GAUC on 30,000 samples.

For MNIST, $GOOD_{100}$ has an excellent GAUC for the training out-distribution 80M Tiny images as well as for notMNIST. For Omniglot, $GOOD_{100}$ is again better than OE/CEDA (similar to EMNIST) in terms of clean AUC's but here ACET is slightly better. However, again it is very difficult to provide any guarantees for this dataset even though non-trival adversarial AUC's against the employed attacks are maintained.

For SVHN, the detection of smooth noise turns out to be the most difficult of the evaluated tasks. There, the clean AUCs of all methods except ACET and CCU are lower than the perfect scores we see on other out-distributions but still very high, and only the higher Quantile GOOD models can give some guarantees. An explanation might be that the image features of SVHN house numbers and of this kind of synthetic noise are similarly smooth. For 80M Tiny Images and Imagenet-, on the other hand, the SVHN high quantile GOOD models, particularly $GOOD_{100}$, are able to provide almost perfect guaranteed AUCs.

For CIFAR-10, on all three out-distributions we again observe the trade-off between clean and guaranteed AUC that comes with the choice of the loss quantile. Overall, the $GOOD_{80}$ model again retains reasonable AUC values for the clean data while also providing useful guaranteed AUCs.

# I Generalization of provable confidence bounds to a larger radius

In Table 9, we evaluate the generalization of empirical worst case and guaranteed upper bound for the confidence within a larger $l_\infty$-ball around OOD samples than what the model was trained for.

As expected, the adversarial AUC's (AAUC) degrade for the larger radius $\epsilon$ for all methods. However, ACET and the GOOD models with higher quantiles maintain their performance much better. Interestingly, while ACET has for the smaller radii typically better AAUCs this is reversed for the larger radii where now often the GOOD models are better, showing that our certified methods can in this aspect sometimes outperform the "adversarial training" approach when it of generalization to higher radii.

On MNIST, $GOOD_{100}$ not only still has a perfect guaranteed AUC for uniform noise for an $\epsilon$ of 0.4 but even on FashionMNIST and CIFAR-10 it still has substantial guarantees.

For SVHN, the excellent guarantees of $GOOD_{100}$ for $\epsilon = 0.03$ generalize well to the doubled radius of $\epsilon = 0.06$ but the gap between GAUC and AAUC increases quite significantly, except for uniform noise where the GAUC is still high at $94.7\%$

For CIFAR-10, even when tripling the evaluation radius to $\epsilon = 0.03$, the certified the bounds of $GOOD_{80}$ generalize surprisingly well: for all out-distributions, we only see an at most moderate drop of the GAUC value compared to Table 1.

In summary, GOOD in most cases still achieves reasonable guarantees for the larger threat model at test time. Moreover, the AAUC for the GOOD models is in most cases better than that of ACET and thus our guaranteed IBP training shows in this regard a better generalization to larger evaluation radii than adversarial training on the out-distribution (ACET).

Table 9: Complementing Table 1, an evaluation of the generalization of worst-case OOD detection, that is AAUC and GAUC, for $\epsilon$-values larger than those of the threat models used during training.

IN: MNIST $\quad \epsilon = 0.4$