[Reviews · NeurIPS 2020]

Review 1

Summary and Contributions: UPDATE AFTER DISCUSSION PERIOD: The rebuttal was very strong and addressed my concerns. I raised my score 5 --> 8. Regarding the name, "adversarially robust OOD" would be fine since it's not overclaiming, but I'd suggest something with "certifiable" or "certifying" in the name (for both the paper title and method name), e.g. something along the lines of: "Certifiable ambivalence on OoD examples"... my main objection is just that (unlike for in-distribution, where you only have out-of-sample generalization issues), the OoD examples you train on may not be representative of OoD examples you encounter at test time (which may be from another OoD distribution). This paper uses interval bound propapagation (IBP) to: 1) train a model to be robustly unconfident about examples within an L_infinity norm ball of OOD data-points AND/OR 2) prove that a model has such robustness properties. It contributes: 1) The above idea, which provides an upper bound on the "log confidence" (i.e. the log probability of the class that the model assigns the highest probability). (eqn6) 2) An upper bound of this upper bound, which is simply the maximal upper bound over all (pairwise) logit differences (eqn7) 3) A more practical (to optimize) approximation of this 2nd upper bound, which is approximately logarithmic for large differences in logits and quadratic for small differences. (eqn8) 4) The idea of ignoring the L_infinity perturbations for the hardest OOD examples (which may in fact be highly similar to in-distribution data). (eqn10, "GOOD_k") 5) Strong experimental results (on MNIST, SVHN, and CIFAR-10) that show that this method (GOOD_k) can provide provably lower confidence around OOD data from several datasets that were never seen at training time.

Strengths: I find the empirical results very impressive, and the methodology is sound. I am intrigued by the authors' suggestion that (paraphrasing): "being provably less confident on OOD data might be a good alternative to being provably correct on adversarially perturbed in-distribution examples". While I was skeptical that this kind of provable robustness would generalize to OOD distributions (or even *examples* other than the one(s) the model was exposed to, the experimental results are very strong in this regard. In terms of novelty, the idea is moderately novel and potentially highly significant. While the method seems to be a fairly straightforward application of IBP, there also appear to be some genuine practical hurdles in applying it successfully in this case, which the authors do a good job of addressing.

Weaknesses: 1) The main weakness of the paper is the way it uses the phrase "worst case OOD detection", which is misleading and not discussed rigorously. In fact, as stated in the abstract, this means "worst case" *within the L_infinity balls around some specific OOD examples*. This paper is *not* providing guarantees about *arbitrary* OOD data, which is, to me, what the phrase "worst case OOD detection" sounds like it refers to. Low confidence can only be guaranteed locally around specific outliers. The empirical results suggest that this may be sufficient in practice in many cases, since exposure on (only) examples from Tiny Images helps provide provable levels of robustness on other OOD datasets at test time. Overall, I found the way the method and results are framed and discussed to actually distract from the scientific significance of this empirical result, which I found somewhat surprising. The authors should be more upfront about this, and emphasize that their method could, in principle, not provide any extra robustness to OOD examples from new OOD datasets, and so it is a somewhat remarkable empirical fact that it seems to do so in practice. 2) A similar somewhat lesser, but still significant, issue: throughout this paper, it is taken for granted that confidence of predictions is what will be used to detect OOD examples (see, e.g. 24-27, 69-71). The authors should clarify that using a confidence threshold of a model trained to perform well in-distribution is only one way of performing OOD detection (for example, an obvious alternative is to treat OOD detection as a binary classification problem and train another model on that task). I believe methods based on thresholding confidence are the state of the art in Deep Learning OOD literature at the moment, but they are a method of solving the task, *not* the task itself. These should be very easy to fix, and I will raise my score if the authors fix them. 3) There could also be a bit more analysis done to pull apart whether GOOD is actually providing more robustness than other methods. When other methods, e.g. ACET, have GAUC=0, but AAUC higher than GOOD's GAUC, we don't know if this is because: 1) "lack of robustness": they are actually less robust, but the adversarial search doesn't find examples that violate the guarantee OR 2) "lack of certifiability": they aren't less robust, but can't be verified to be robust using IBP, because they haven't been trained to satisfying IBP-based upper bounds on confidence. It would be nice to try and distinguish 1 and 2. e.g. one obvious question is: are the GAUC scores for other methods still 0 even as epsilon (the size of the norm ball) goes to 0? If so, this suggests that (2) is playing a significant role (since then the robustness that GAUC seeks to guarantee basically reduces to AUC (in Table 1)). 4) The authors should also further justify the exclusion of the 4.25M images (198-200). This appears to be non-standard, and it seems like excluding these images might have been necessary for GOOD to work (else why go through the extra hassle?) So an ablation should be performed, and if it yields significantly different/worse results, the implications and potential limitations revealed should be discussed. I hope the authors address these concerns so I can raise my score. I think the paper is a clear accept if all of these issues are addressed. In terms of their significance as weaknesses, I would order them from most to least important as 1,4,2,3.

Correctness: Yes. This is somewhat notable because methodology is particularly important for OOD data.

Clarity: The paper is very well written and quite clear. As someone who is not intimately familiar with IBP, I would appreciate an explanation of eqn3. The weaknesses of framing I mentioned above also reduce the clarity somewhat, but overall it was still clear enough once I had read the whole thing.

Relation to Prior Work: Yes. The authors clearly explain: 1) That they are using IBP to construct their bound, and how they do so. 2) How their method compares to previous methods such as outlier exposure (OE) and ACET... Essentially, OE encourages low confidence on OOD examples, and ACET uses an adversarial search method to also encourage robustness *around* OOD examples. GOOD instead uses IBP to optimize an upper bound on the confidence of any example within an L_infinity norm ball of an OOD example. The authors also mention (and compare with) Certified Certain Uncertainty (CCU), another method which can prove that some OOD data will be predicted with low confidence, and provide details on its relation to GOOD in the appendix. The methods seem substantively different.

Reproducibility: Yes

Additional Feedback: A few small things: - In the abstract "allowing to trigger" is missing a subject (who is doing the triggering?), and "we are aiming" could be "we aim". - Paragraph 212-224 was a bit hard to read because of the long phrase: "the maximal confidence in the l_infty-ball of radius eps around each out-distribution image" which appears in some variation 3 or 4 times in the paragraph. - Figure 1 is made a bit harder to understand because the methods are only referred to via acronyms and not spelled out or referenced. - Please provide a reference for "gradient masking" (220). - grep "accurcay".


Review 2

Summary and Contributions: The paper addressed the problem of out-of-distribution detection using outlier exposure. Instead of just minimizing the confidence on outlier points as done in prior work, the paper attempts to minimize the worst cast confidence under an \epsilon ball around the input. Borrowing ideas from adversarial robustness literature, the paper proposes using Interval Bound Propagation to provide provable guarantees for OOD detection under the additive perturbation model. Experimental results show that the method achieves good OOD scores comparable to SOTA OOD detection approaches in addition to providing provable guarantees for perturbed OOD samples within the \epsilon ball.

Strengths: The idea of using IBP for provable OOD detection seems interesting. One of the shortcomings of using IBP for adversarial robustness in low natural accuracy. In this paper, the authors show that it is not the case for OOD detection. It is possible to train models with IBP and retain AUC scores on unperturbed samples as that of other SOTA OOD methods, which I found very interesting. And these results are obtained for deeper nets (5Conv + 3FC layer nets), which is nice. The experiments are extensively performed. For multiple in-distribution and OOD datasets, AUC, AAUC and GAUC are reported. And their method seems to perform well across all settings. Overall, I like the paper and enjoyed reading it.

Weaknesses: The loss function where quadratic loss is used inside log function (Eq. 8) seems a bit adhoc. Why use this particular choice of loss function. Why should it be quadratic and not higher order polynomials? Was any other choice of loss fns considered? Can you comment if the method could be used on deeper models like Resnets? Evaulation for AAUC is performed with \epsilon=0.01 for SVHN/CIFAR models which essentially translates to a 2-pixel perturbation. What happens when you use larger perturbation balls. The reason why I bring this up is because for OOD inputs, you can potentially use much larger \epsilon balls and the models should still classify them as OOD. Here, we don't care about imperceptible perturbations. In most cases, you would need really large perturbation balls to make OOD inputs look like in-distribution points. Hence, for all these perturbations, an ideal model should classify as OOD. While perturbations are crafted with the objective of making OOD as in-distribution, what happens in the other direction -- adding small perturbations to make in-distribution points predict as OOD. In this case, IBP can give you a bound until which in-distribution points still remain in-distribution.

Correctness: The empirical validation seems to be consistent with prior evaluations.

Clarity: The paper is very well written and easy to follow. I enjoyed reading it.

Relation to Prior Work: Relation to prior work is also explained well.

Reproducibility: Yes

Additional Feedback: Instead of training GOOD models for different quantiles q, is it possible to use a curriculum for q on the fly. One way to do this would be begin with q=0, and progressively increasing it as the model trains.


Review 3

Summary and Contributions: This paper derives a new Confidence Upper Bound based on IBP and applies it to OOD. Experiments show the proposed method can achieve pretty good OOD performance, especially for GAUC.

Strengths: 1. The derivation of the upper bound / L_CUB is clear. The theoretical result may also have some positive effects in the field of OOD. 2. The paper is well-written.

Weaknesses: 1. Confusion about Fig. 1. Does GOOD_80/ACET give low confidence about all categories? Although GOOD_80/ACET give low confidence about dog, it is meaningless if they give high confidence about another class, like cat. What is the whole distribution of the output of GOOD_80/ACET? 2. According to Sec.3 and Line174-182, I think it is pretty hard to apply GOOD in a real problem. Many hyper-parameters need to be finetuned. Besides, Line130-132 "as the bound can be several orders of magnitude larger than the in-distribution loss. Therefore, we use the logarithm of this quantity." I think the magnitude doesn't matter, but the gradient of the loss matters. Can authors provide some quantitative analysis about why they apply logarithm to Eq.(7)? 3. In Tab.1, ACET/CCU/OE get 0.0 GAUC, is this because of their intrinsic methodology and cannot be easily modified? Why GOOD achieves non-zero GAUC while others cannot? 4. In Tab.1, GOOD does not show much improvement than ACET/CCU/OE in terms of AUC/AAUC. On CIFAR-10, it gets much worse results than ACET/CCU/OE in terms of AUC/AAUC. However, the results of AUC/AAUC is comparable on MNIST/SVHN. It is pretty weird. Is the performance of GOOD related to the dataset?

Correctness: Correct.

Clarity: Yes.

Relation to Prior Work: Yes.

Reproducibility: Yes

Additional Feedback: The authors address my concerns in the rebuttal.

[Author Response · NeurIPS 2020]

We thank all reviewers for their helpful feedback. All mentioned minor issues will be fixed in the final version. We
appreciate that all reviewers like the idea of certifiable adversarially robust OOD detection and that they are as positively
surprised as we were that this is possible with only minor or no loss in test accuracy.

**R1: Ambiguity of 'worst case OOD detection'.** We agree that the term 'worst case OOD detection' is ambiguous
and propose to use 'adversarially robust OOD detection' but we are open for suggestions. The generalization of our
guarantees from OOD training to different OOD test sets is indeed an empirical observation and cannot be proven (same
for certified robustness on the in-distribution) and we clarify this in the final version.

**R1: Confidence thresholding for OOD detection.** We modify the relevant sections to highlight that other approaches
rather than confidence thresholding exist for OOD detection.

**R1/3: GAUC for ACET and other methods.** We do not know whether the true values of the WC-AUC lie closer to
the GAUC (lower bound) or the AAUC (upper bound) for the different methods, but we believe that both bounds are not
tight. We tested for the largest epsilon for $\epsilon = 10^{-k}$ where IBP gives non-zero GAUCs. This is $k = 6$ (ACET), $k = 7$
(OE) and $k > 7$ (Plain). This provides weak evidence that a more robust model is also more verifiable. On the other
hand this phenomenon that adversarially trained models are not certifiable is well-known for adversarial in-distribution
robustness. However, for adversarial training on the in-distribution it is generally believed that the empirical robustness
is close to the true one. In our case we don't think so as adversarial attacks on ACET (and sometimes even OE/CEDA)
are much more difficult as the gradients are often very small or even zero and thus some of the attacks might simply fail
even though the point is not robust. We discuss these difficulties in Appendix A. For the final version we try to adapt
Mixed-Integer-Programming for certification to our setting to further investigate this question.

**R1: Pre-filtering of Tiny-Image Dataset as OOD training distribution.** The pre-filtering step is an artefact of our
initial attempts to stabilize the IBP training which we later on did not challenge again - so thanks a lot for this question.
In fact the inclusion of the 4.25M discarded images into the training set does not change any of our results (very minor
positive and negative changes with no trend). Thus, for the final version we will report all results for the full TinyImage
dataset and only exclude CIFAR in order to be comparable to the setting of Outlier exposure (OE) [19] (we can even
include the 132K CIFAR images into training - for our quantile-based loss it is no problem if there is a small overlap of
in- and out-distribution as it was exactly designed for this - but then we could not use CIFAR-10/100 for evaluation).

**R2/3: Form of 'Confidence Upper Bound Loss' $\mathcal{L}_{\text{CUB}}$ (line 134).** The logarithm also has an effect on the gradient:
it leads for large upper bounds roughly to a rescaling of the gradient of the upper bound on the confidences by the
actual upper bound and thus is essential for a better behaved training. The square inside the loss was chosen as this
leads to more uniformly small upper bounds over all OOD images compared to not squaring (similar to $l_2$- vs $l_1$-loss).
Higher order polynomials would not be of additional help here and might lead to numerical problems. We found that
the omission of either the $\log$ or the squaring, or both, makes training less stable.

**R2: Train deeper architectures e.g. ResNets.** We have adapted our IBP training to a Fixup ResNet-20 without BN
and Dropout. Interestingly it is possible to train these deeper models and obtain similar but worse GAUCs than with
our XL architecture. Since the worse results might be due to an suboptimal schedule or initialization, we are currently
trying to identify and resolve the cause of this performance gap.

**R2: Larger $\epsilon$-balls.** Note that we evaluate AAUC/GAUC of our models at larger radii in Appendix I (Table 9) and the
guarantees partially generalize. For the rebuttal we trained models on CIFAR-10/SVHN with $\epsilon = 8/255$. On CIFAR-10
$\text{GOOD}_{80}$ has an accuracy of 90.6 (notably accuracy is still unaffected) and achieves clean AUCs of 78.6/95.0/89.9/96.2
(CIFAR-100/SVHN/LSUN/Uni) vs. 85.9/95.6/96.2/90.4 for the $\text{GOOD}_{80}$ model trained with $\epsilon = 0.01$ and GAUCs at
$\epsilon = 8/255$ of 43.2/17.9/51.1/94.0 vs. 36.7/34.4/48.4/86.9 so the resulting changes appear inconclusive. For SVHN
the differences are stronger. The $\text{GOOD}_{100}$ model trained for $\epsilon = 8/255$ achieves an accuracy of 96.0 (vs. the 96.6 of
$\text{GOOD}_{100}$ trained with $\epsilon = 0.01$) and has clean AUCs of 99.5/99.7/99.9/100 (CIFAR-100/CIFAR-10/LSUN/Uni) vs.
99.9/100/100/100 as well as GAUCs for $\epsilon = 8/255$ of 96.0/97.3/98.5/100 vs. only 40.3/41.3/40.3/1.5 for $\text{GOOD}_{100}$
trained with $\epsilon = 0.01$. This confirms the intuition of R2 that for SVHN a larger $\epsilon$ on the OD is feasible. We include all
results in the final version and explore even larger radii.

**R2: Perturbations on in-distribution inputs.** We used IBP to compute lower bounds on the confidence in the original
class around in-distribution points and got non-trivial certificates only for very small radii. Note that certified adversarial
robustness on the in-distribution [13] comes with a larger drop in test accuracy which we want to avoid.

**R2: On the fly curriculum for the quantile $q$ in GOOD.** Thanks, we thought about this but discarded it in favor of
an ablation study on the effect of the quantile-loss, but it is definitely an interesting direction to pursue.

**R3: Definition Confidence/Clarification of Fig. 1.** In line 67, we define confidence of an input $x$ as the maximum of
the predicted probability distribution over the classes, which for all models happens to be realized by *dog* in all cases.
Thus the probabilities of all other classes are lower than the one of *dog*.

**R3: Effect of GOOD on AUC on MNIST/SVHN vs CIFAR-10.** Since CIFAR-10 is a subset of TinyImages and
thus in- and training out-distribution are more similar, the task of provable OOD detection is significantly harder for
CIFAR-10 than for MNIST/SVHN which affects the clean AUC (see lines 261-267). Since ACET directly optimizes
empirical robustness, it unsurprisingly tends to have good AAUCs. For discussion of ACET/CCU see Appendix B.

[Meta-Review · NeurIPS 2020]

Reviewers found the idea of the paper interesting and authors did a good job in addressing the comments in their responses. In the final draft, authors should explain clearly limitations of their results identified in the rebuttal (inability to train with deeper models etc).